# A multi-hierarchical approach reveals D-serine as a hidden substrate of sodium-coupled monocarboxylate transporters

**Pattama Wiriyasermkul[1,2,3†], Satomi Moriyama[3], Masataka Suzuki[4], Pornparn Kongpracha[1,2], Nodoka Nakamae[3], Saki Takeshita[3], Yoko Tanaka[3], Akina Matsuda[4], Masaki Miyasaka[1,2], Kenji Hamase[5], Tomonori Kimura[6,7], Masashi Mita[8], Jumpei Sasabe[4], Shushi Nagamori[1,2,3]\***

[1]Center for SI Medical Research, The Jikei University School of Medicine, Tokyo, Japan; [2]Department of Laboratory Medicine, The Jikei University School of Medicine, Tokyo, Japan; [3]Department of Collaborative Research for Biomolecular Dynamics, Nara Medical University, Nara, Japan; [4]Department of Pharmacology, Keio University School of Medicine, Tokyo, Japan; [5]Graduate School of Pharmaceutical Sciences, Kyushu University, Fukuoka, Japan; [6]KAGAMI Project, National Institutes of Biomedical Innovation, Health and Nutrition, Osaka, Japan; [7]Reverse Translational Research Project, Center for Rare Disease Research, National Institutes of Biomedical Innovation, Health and Nutrition, Osaka, Japan; [8]KAGAMI Inc, Osaka, Japan

**\*For correspondence:**
snagamori@nagamori-lab.jp

**Present address:** [†]Department of Biological Chemistry and Food Science, Faculty of Agriculture, Iwate University, Iwate, Japan

**Abstract** Transporter research primarily relies on the canonical substrates of well-established transporters. This approach has limitations when studying transporters for the low-abundant micro-molecules, such as micronutrients, and may not reveal physiological functions of the transporters. While D-serine, a trace enantiomer of serine in the circulation, was discovered as an emerging biomarker of kidney function, its transport mechanisms in the periphery remain unknown. Here, using a multi-hierarchical approach from body fluids to molecules, combining multi-omics, cell-free synthetic biochemistry, and ex vivo transport analyses, we have identified two types of renal D-serine transport systems. We revealed that the small amino acid transporter ASCT2 serves as a D-serine transporter previously uncharacterized in the kidney and discovered D-serine as a non-canonical substrate of the sodium-coupled monocarboxylate transporters (SMCTs). These two systems are physiologically complementary, but ASCT2 dominates the role in the pathological condition. Our findings not only shed light on renal D-serine transport, but also clarify the importance of non-canonical substrate transport. This study provides a framework for investigating multiple transport systems of various trace micromolecules under physiological conditions and in multifactorial diseases.

## eLife assessment

This study shows **compelling** evidence that the less common D-serine stereoisomer is transported in the kidney by the neutral amino acid transporter ASCT2 and that it is a non-canonical substrate for sodium-coupled monocarboxylate transporter SMCTs. With a multi-hierarchical approach, this **important** study further shows that Ischemia-Reperfusion Injury in the kidney causes a specific increment in renal reabsorption carried out, in part, by ASCT2.

## Introduction

Membrane transport proteins function to transport micromolecules such as nutrients, ions, and metabolites across membranes, thereby playing a pivotal role in the regulation of micromolecular homeostasis. To date, over 450 solute carriers (SLCs), 40 ATP-binding cassettes, 60 ATPase pumps, and 300 ion channels have been identified in mammals (*Ranjan et al., 2011*; *César-Razquin et al., 2015*; *Perland and Fredriksson, 2017*; *Bateman et al., 2021*). Approximately 30% of membrane transport proteins remain uncharacterized. While some membrane transport proteins can be classified based on sequence similarities and transport substrates of well-studied family members, this approach is limited when it comes to discovering transporters for some micromolecules, such as micronutrients and their metabolites. Many of these micromolecules have not been extensively studied and do not fit into traditional nutrient types. Currently, the importance of micronutrients and gut microbiota-derived metabolites in health and disease has gained a high impact (*Shenkin, 2006*; *Vernocchi et al., 2016*; *Liu et al., 2022*), yet their responsible transporters are not well understood. Understanding the absorption and reabsorption mechanisms of dietary micronutrients and gut microbiota-derived metabolites will provide a key molecular process for controlling their dynamics and molecular activities.

This study aims to establish an approach to investigate the physiological function and pathological significance of transporters, particularly transport systems for trace micromolecules. As a paradigm, we investigated renal transport systems for D-amino acids, micronutrients derived mostly from the diet, and some from the gut microbiota. Amino acid transport systems have been well studied (*Bröer, 2008a*), but have focused on L-amino acids, which are macronutrients, leaving open the question of how D-amino acids are transported. We have developed two-dimensional high-performance liquid chromatography (2D-HPLC), which allows the segregation of L- and D-enantiomers in liquid biopsy (*Miyoshi et al., 2009*; *Hamase et al., 2010*). The importance of D-amino acids in renal function has received increasing attention (*Sasabe et al., 2014*; *Kimura et al., 2016*; *Sasabe and Suzuki, 2019*; *Inoue et al., 2018*; *Hesaka et al., 2019*; *Kimura et al., 2020*; *Okushima et al., 2021*; *Iwata et al., 2022*; *Suzuki et al., 2022*). D-Serine in the body fluid was discovered as a promising biomarker for acute kidney injury (AKI) and chronic kidney disease (CKD) due to its clear association with kidney function in rodents and humans (*Sasabe et al., 2014*; *Kimura et al., 2016*; *Inoue et al., 2018*; *Hesaka et al., 2019*; *Okushima et al., 2021*). Diseased-associated alterations in plasma and urinary D-/L-serine ratios suggest that renal D-serine transport systems are different and distinct from L-serine transport systems under physiological and pathological conditions. The transport of D-serine has long been found to take place at the proximal tubules (*Kragh-Hansen and Sheikh, 1984*; *Shimomura et al., 1988*; *Silbernagl et al., 1999*; *Sasabe and Suzuki, 2019*), yet the corresponding transporter has not been clearly elucidated.

D-Serine is found at submillimolar levels in the brain (*Hashimoto et al., 1992*), where it acts as an obligatory physiological co-agonist of *N*-methyl-D-aspartate receptors (*Mothet et al., 2000*; *Basu et al., 2009*; *Papouin et al., 2012*; *Wolosker, 2018*). Extensive studies of D-serine in the brain show that D-serine is transported by four plasma membrane transporters: ASCT1 (SLC1A4), ASCT2 (SLC1A5), Asc-1 (SLC7A10), and SNAT1 (SLC38A1) (*Rosenberg et al., 2013*; *Foster et al., 2016*; *Kaplan et al., 2018*; *Bodner et al., 2020*). In contrast, D-serine is detected at low micromolar levels in the periphery, and little is known about its dynamics as well as transport systems. Mammals acquire D-serine by biosynthesis via serine racemase function (*Wolosker et al., 1999*) and absorption from the diet and gut microbiota presumably via intestinal transport system(s) (*Sasabe and Suzuki, 2019*; *Inoue et al., 2018*; *Gonda et al., 2023*). Similar to the renal epithelia, a D-serine transporter in the intestine is also unknown.

In this study, we integrated multiple analytical methods for the different biological hierarchies. We applied 2D-HPLC to quantify the plasma and urinary enantiomers of 20 amino acids of renal ischemia-reperfusion injury (IRI) mice, a model of AKI and AKI-to-CKD transition (*Sasabe et al., 2014*; *Fu et al., 2018*). Membrane proteomics of the renal proximal tubules of the IRI mouse models was performed to explore the key molecules responsible for the alterations of D- and L-amino acid transport. With bioinformatics, cell-based screening, and cell-free transport analysis, we identified two types of D-serine transport systems in the kidney. Subsequently, ex vivo transport analyses from the normal kidney and the IRI model explained the transport mechanism of D-serine as a biomarker in kidney diseases.

# Results

## Amino acid metabolomics in plasma and urine of the renal IRI model

To understand kidney-associated alteration of enantiomeric amino acids, we measured L- and D-isomers of 20 amino acids in the body fluids of the IRI model (*Figure 1—figure supplements 1–2*). Dynamics of L- and D-enantiomers in the plasma were evaluated from the ratios of D-/L- enantiomers (*Figure 1A*). Among amino acids, only the ratios of plasma D-/L-serine were increased in a time-dependent manner from 4 to 40 hr after the ischemia-reperfusion (*Figure 1A and B*). The elevation of plasma D-/L-serine ratios at early time points (4 hr–8 hr) was due to a sharp decrease of the L-isomer, while the rise of D-/L--serine ratios at late time points (20 hr–40 hr) was a result of a continuous acceleration of the D-isomer (*Figure 1C and D*). The results of plasma D-serine were consistent with previous observations (*Sasabe et al., 2014*). In addition to D-serine, we detected certain amounts of D-alanine, D-proline, and D-methionine in the plasma (*Figure 1—figure supplement 1A*). However, their D-/L- enantiomeric ratios were uncorrelated to the pathological conditions (*Figure 1A*). Our knowledge of traditional amino acid transport suggests that alanine, serine, and proline share the same transport systems (*Bröer, 2008a*). The noticeably different profile of plasma D- and L-serine from those of alanine and proline led to the speculation that D-serine transport was possibly mediated by unique transport systems, different from those of D-alanine, D-proline, and the L-isomers. Notably, such transport systems would be sensitive to the IRI conditions.

We measured the enantiomeric amino acid profiles from urine samples of the IRI model (*Figure 1—figure supplement 2*). While plasma is a reliable resource of metabolite biomarkers, urine is a meaningful non-invasive diagnostic indicator of kidney function. More importantly, the analysis of urinary metabolites is indispensable for elucidating the unknown reabsorption systems of certain substances in the kidney. From our results, the ratios of urinary D-/L-serine were drastically decreased at early IRI (4 hr–8 hr) (*Figure 1E–F*). Similar tendencies of D-/L-amino acid ratios were observed in alanine, proline, asparagine, and histidine (*Figure 1E–H*). These urinary amino acid profiles indicated the impairment in both D- and L-amino acid reabsorption since the early injury. We hypothesized that transporters play a major role in the dynamics of D- and L-serine in the kidney with respect to changes in the D-/L-serine ratio during pathology.

## Proteomics for membrane transport proteins from renal brush border membranes of the IRI model

We aimed to identify the corresponding molecules for the dynamics of D- and L-serine. To obtain the candidates, we performed proteomic analysis of renal brush border membrane vesicles (BBMVs; the membrane fraction enriching apical membrane proteins of proximal tubular epithelia) from the IRI model. We analyzed the BBMVs from 4 to 8 hr after IRI, the times of rigorous shift of plasma and urinary D-/L-serine ratios (*Figure 1B and F*). The proteomes of the IRI samples were calculated as ratios of the negative control group (sham), yielding two groups of comparative proteome data: 4 hr IRI/sham (4 hr) and 8 hr IRI/sham (8 hr). In total, 4423 proteins were identified, of which 1187 proteins (27% of the total detected proteins) were categorized as plasma membrane proteins and extracellular matrix proteins (*Supplementary file 1*). We observed a significant increase of two well-known early AKI biomarkers: Ccn1 (Cyr61: cellular communication network factor 1; 4 hr IRI/sham = 41.4 folds; 8 hr IRI/sham = 29.9 folds) and Ngal (Lcn2: neutrophil gelatinase-associated lipocalin; 4 hr IRI/sham = 1.8 folds; 8 hr IRI/sham = 9.5 folds). This finding confirmed the reliability of our proteomics for molecular analysis of early AKI (*Supplementary file 1*; *Marx et al., 2018*).

From the proteome, we detected 398 membrane transport proteins (325 transporters and 73 ion channels). Analysis of all membrane transport proteins by Ingenuity Pathway Analysis (IPA) revealed alterations of the biological function in *Molecular Transport*, which are categorized by the types of their canonical substrates (*Supplementary file 2*). Transport of 'all micromolecules' was shown to be drastically decreased from 4 hr IRI (*Figure 2A*: top). This decrease was due to the dramatic reduction of transport systems for heavy metals and organic compounds (*Figure 2A*). *Figure 2B* shows membrane transport proteins corresponding to the annotation in *Figure 2A*. This proteomic result unveiled the whole picture of the molecular targets of the injury. Furthermore, the results revealed the key membrane transport proteins behind AKI metabolite biomarkers. It is noted that some transporters were upregulated, suggesting the compensatory-mechanism repair processes for cell survival from the injury (*Figure 2B*).

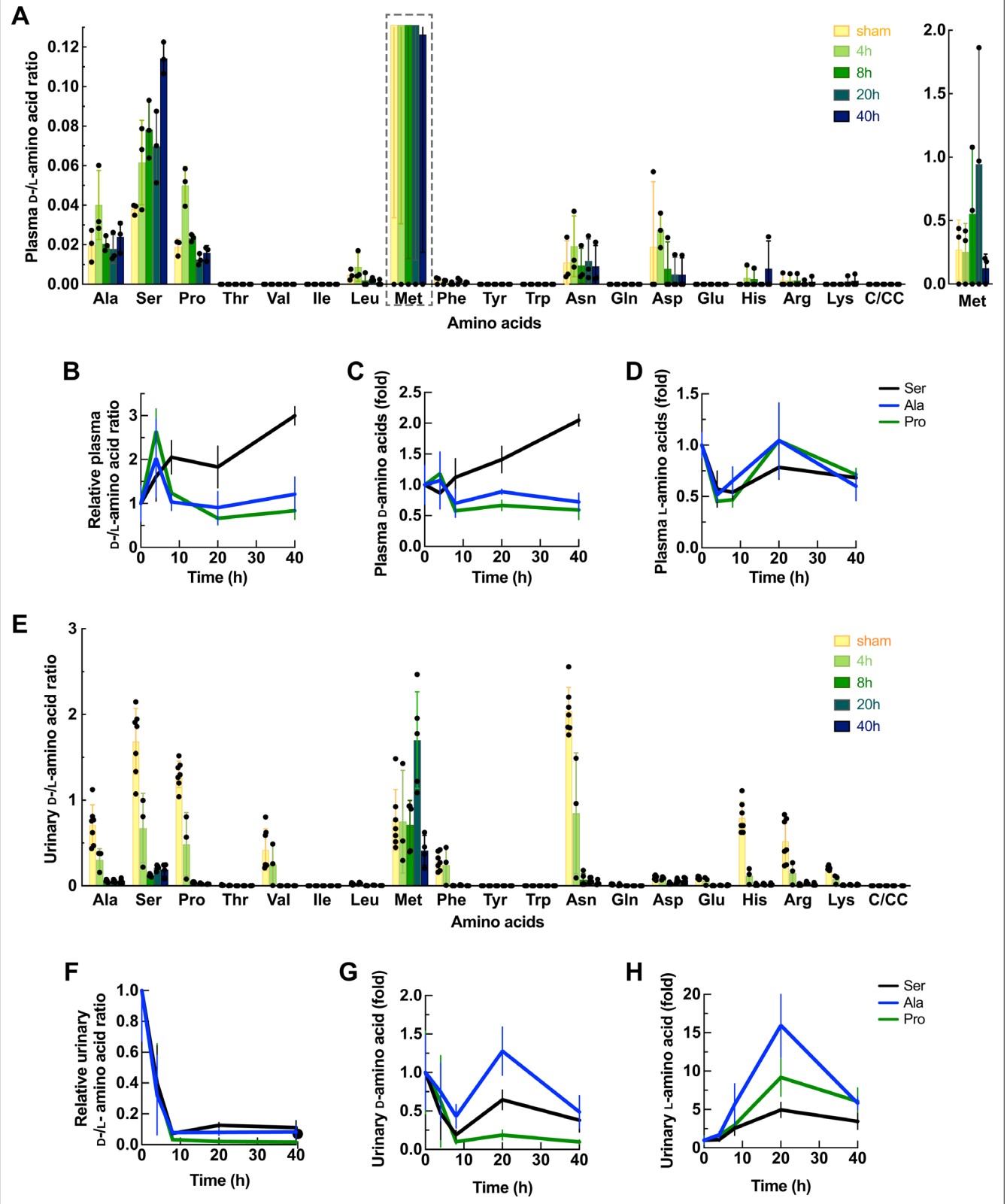

**Figure 1.** Enantiomeric profiles of D- and L-amino acids in plasma and urine of the ischemia-reperfusion injury (IRI) model. Plasma and urine were collected from the mice after ischemia operation for 4, 8, 20, and 40 hr or sham operation (0 hr). The concentrations of 20 amino acids were measured by two-dimensional high-performance liquid chromatography (2D-HPLC) and plotted as mean ± SD. (**A**) Ratio of D-/L-amino acids from the plasma of the IRI model. The graph of Met is shown separately. n=3. (**B–D**) Plasma amino acid profiles of serine, alanine, and proline from (**A**) were normalized with 0 hr

*Figure 1 continued on next page*

*Figure 1 continued*

and plotted as ratios of D-/L- enantiomers (**B**), relative concentrations of D-isomers (**C**), and relative concentrations of L-isomers (**D**). (**E**) Ratio of D-/L-amino acids from the urine of the IRI model. n=3–7. (**F–H**) Urinary amino acid profiles of serine, alanine, and proline from (**E**) were normalized with those at 0 hr and plotted as ratios of D-/L- enantiomers (**F**), relative concentrations of D-isomers (**G**), and relative concentrations of L-isomers (**H**). C/CC: cysteine or cystine.

The online version of this article includes the following figure supplement(s) for figure 1:

**Figure supplement 1.** Enantiomeric profiles of D- and L-amino acids in plasma of the ischemia-reperfusion injury (IRI) model.

**Figure supplement 2.** Enantiomeric profiles of D- and L-amino acids in urine of the ischemia-reperfusion injury (IRI) model.

## ASCT2 is one of D-serine transporters at the apical membrane of renal proximal tubules

Given insight into transporters in the proteome, we observed the elevation of mouse Slc1a5/Asct2, a known D-serine transporter in the brain, at both 4 hr and 8 hr IRI (*Figure 2B*: transport of amino acids). Asct2 was previously detected in renal brush border membranes, but its precise localization and function in the kidney have not been characterized (*Avissar et al., 2001*; *Scalise et al., 2018*). We then examined the localization of Asct2 in the kidney by using affinity-purified Asct2 antibodies that recognized its N-terminus (NT) or C-terminus (CT) (*Figure 3—figure supplement 1A*). Asct2 was partially co-immunostained with both Sglt2 (Slc5a2; sodium/glucose cotransporter 2) and Agt1 (Slc7a13, aspartate/glutamate transporter 1), which are apical membrane markers for S1+S2 and S3 segments, respectively (*Figure 3A and B*; *Nagamori et al., 2016a*; *Ghezzi et al., 2018*). In contrast, Asct2 did not co-localize with Na⁺/K⁺-ATPase, a basolateral membrane marker (*Figure 3C*). The results demonstrated that Asct2 is localized at the apical side in all segments of proximal tubules.

To confirm the D-serine transport function of ASCT2 in human cells, we used wild-type (WT) near-haploid human leukemic cancer cell line (HAP1) and HAP1 cells carrying CRISPR/CAS-mediated *ASCT2* knockout (ASCT2-KO). D-Serine uptake was significantly decreased in the ASCT2-KO cells (*Figure 3D*). The function of ASCT2 was also verified in human embryonic kidney HEK293 cells, which endogenously express ASCT2, by using ASCT2 knockdown (*Figure 3E*).

## Screening of candidate molecules for D-serine transporters

While we have shown that ASCT2 is a D-serine transporter at the apical membranes of all proximal tubular segments, ASCT2 transports D-serine with high affinity ($K_m$ of 167 μM in oocyte system) but with weak stereoselectivity (*Foster et al., 2016*). These properties of ASCT2 differ from those of the reported D-serine transport systems in the kidney. First, D-serine transport systems in S1+S2 segments are reported to be different from those in the S3 segment and the kinetics of both systems are in mM range (*Kragh-Hansen and Sheikh, 1984*; *Silbernagl et al., 1999*). Silbernagl et al. also suggested that ASCT2 is not (or not only) a D-serine transporter at S3 segment (*Silbernagl et al., 1999*). Second, the plasma and urinary D-/L-serine profiles from CKD and AKI samples indicated the stereoselectivity of the transporter (*Figure 1*; *Sasabe and Suzuki, 2019*). Third, ASCT2 recognizes serine, alanine, and proline, but our enantiomeric amino acid profile in plasma samples showed a distinct D-/L-ratio pattern of serine from alanine and proline (*Figure 1B*). All data suggest the existence of (an)other D-serine transporter(s).

To search for the D-serine transporters, we chose the ratios of mouse Asct2 as the cutoff values of our membrane proteomic data at both 4 and 8 hr IRI (*Figure 4A*). Membrane transport proteins showing more alteration than Asct2 at both increased and decreased values were selected because some transporters are known to switch roles upon pathology. Further, we focused on the luminal transporters and omitted transporters that are reported to be located at basolateral membranes and organelles as the BBMVs are enriched apical membrane fractions. Membrane transport proteins that recognize only inorganic ions were also excluded. Finally, 10 candidates were selected (*Table 1*). Slc36a1/Pat1 and Slc6a18/B⁰at3, the known small amino acid transporters, were included in the list, although their expressions only passed the cutoff at only one time point (either 4 or 8 hr IRI). Slc5a12/Smct2 was also included because Smct1, another member of sodium-coupled monocarboxylate transporter (SMCT) family, was selected as a candidate. Smct2 has a comparable role with Smct1, but both transporters are localized at different segments of the proximal tubules.

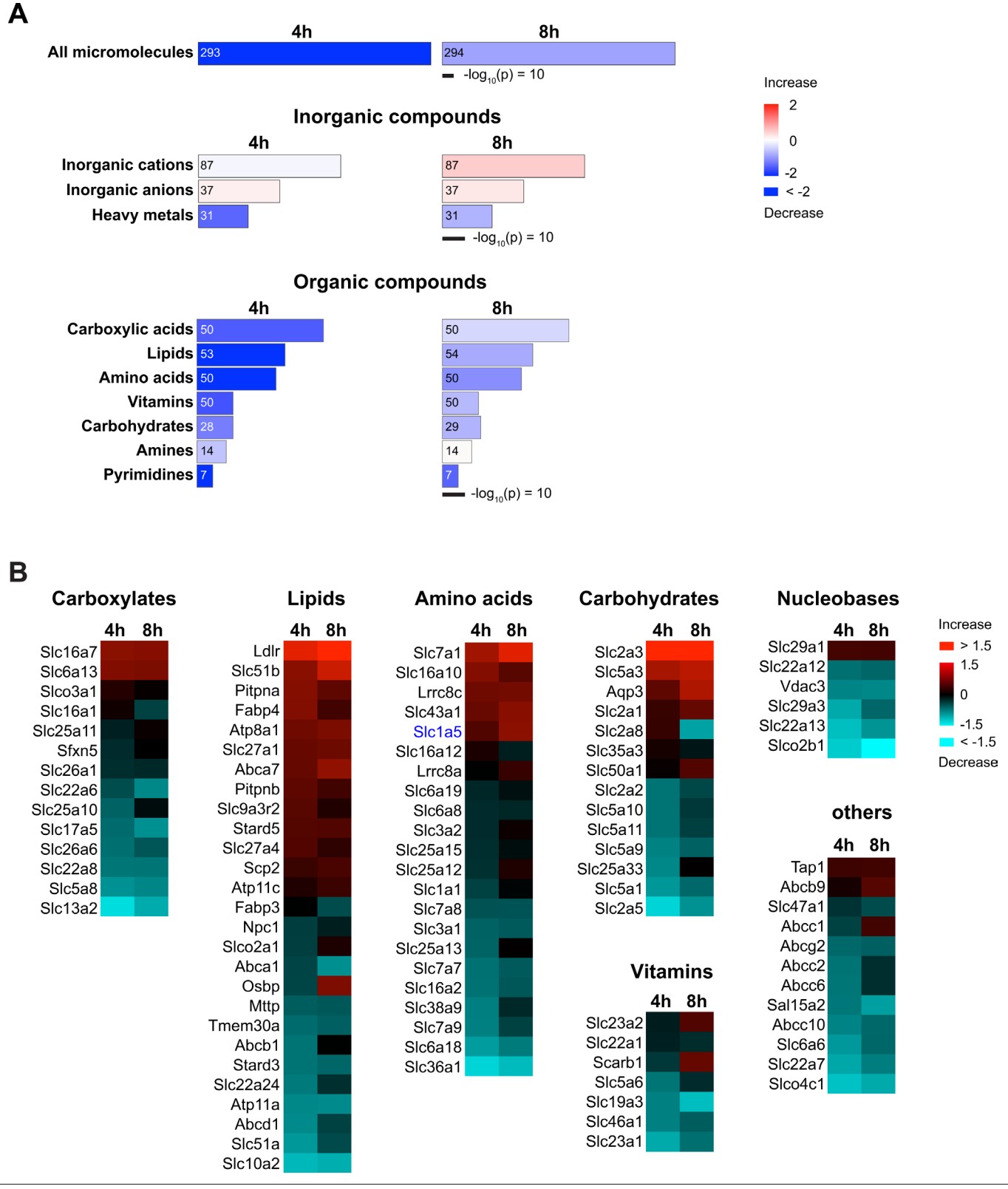

**Figure 2.** *Molecular Transport* in renal brush border membrane vesicles (BBMVs) proteome of the ischemia-reperfusion injury (IRI) model. (**A**) Ingenuity Pathway Analysis (IPA) shows the heatmaps of *Molecular Transport* in the membrane proteome from BBMVs of the IRI model (4 hr IRI/sham [4 hr] or 8 hr IRI/sham [8 hr]). Transport functions are categorized by types of substrates. Transport of all micromolecules (top) is derived from the combination of inorganic (middle) and organic compounds (bottom). Area and colors represent –$\log_{10}$(p-value) and annotated functions (z-score), respectively. Numbers

*Figure 2 continued on next page*

Figure 2 continued

inside the columns indicate the numbers of corresponding proteins. (**B**) Heatmaps of membrane transport proteins that mediate the transport of organic compounds. The proteins with ratios of more than 1.1-fold change are shown. The category 'others' includes the transport of peptides, organic cations, organic anions, and drugs. Colors indicate $\log_2$ fold of 4 hr IRI/sham (4 hr) or 8 hr IRI/sham (8 hr).

To analyze the function of the candidate transporters, we selected HEK293 cell line for the screening method because of its high transfection efficiency. Similar to other several common cell lines, HEK293 cells express ASCT2 endogenously. However, we also examined other possible D-serine transport systems in HEK293 cells, aiming to reduce the transport background. Proteomic analysis of the membrane fractions from HEK293 cells identified amino acid transporters as shown in ***Supplementary file 3***. Transport activity of D-serine in HEK293 cells showed Na+ dependency and ASCT2 substrate-mediated inhibition (L-Ser, L-Thr, and L-Met) (***Figure 3—figure supplement 1B and C***). Ben-Cys (S-benzyl-L-cysteine) and GPNA (L-γ-glutamyl-p-nitroanilide) inhibited D-serine transport in HEK293 cells similar to that of ASCT2-expressing cells (***Bröer et al., 2016***), but SLC38 inhibitor MeAIB (2-(methylamino)isobutyric acid) had no effect (***Figure 3—figure supplement 1C***). These results indicated that ASCT2 is the main D-serine transporter in HEK293 cells.

Due to the absence of any ASCT2-specific inhibitor and the compensatory transport mechanism that occurred in the ASCT2-deletion condition (***Bröer et al., 2016***), we avoided using the traditional cell-based transport assay for screening. Instead, we established a new screening method based on cell growth. Apart from the studies on D-serine biomarkers in kidney diseases, a previous study found that high concentrations (10–20 mM) of D-serine impaired the growth of a human proximal tubular cell line (***Okada et al., 2017***). We then tested whether high concentrations of D-serine affect HEK293 cell growth. Unlike L-serine which had no effect, D-serine reduced cell growth in a concentration-dependent manner with $IC_{50}$ of 17.4±1.05 mM (***Figure 3—figure supplement 1D***). In the ASCT2 knockdown cells, the growth inhibition by D-serine was attenuated (***Figure 4B***), indicating that D-serine suppressing cell growth was caused by ASCT2-mediated D-serine transport.

Based on this finding, we utilized cell growth determination assay as the screening method even in the presence of endogenous ASCT2 expression. HEK293 cells were transfected with human candidate genes without ASCT2 knockdown. The transfected cells were treated with either 15 mM (near $IC_{50}$ concentration) or 25 mM (high concentration) D-serine for 2 days, and cell growth was determined. The positive control Asc-1-transfected cells showed a reduction in cell growth, confirming the effectiveness of this assay (***Figure 4C and D***). Among 10 candidates, PAT1 (SLC36A1) and B0AT3 (SLC6A18) showed higher cell growth than the mock cells, suggesting that PAT1 and B0AT3 may reduce D-serine influx mediated by endogenous ASCT2. In contrast, SMCT1 (SLC5A8) and SMCT2 (SLC5A12) extended the growth suppression after treatment with both 15 and 25 mM D-serine (***Figure 4C and D***, ***Figure 4—figure supplement 1A and B***). Thus, with the hypothesis that the candidate molecule influxes D-serine and brings about growth reduction, we selected both SMCT1 and SMCT2 for further analysis.

## Characterization of D-serine transport in SMCTs

SMCTs are known to transport monocarboxylates such as lactate, propionate, and nicotinate in a Na+-dependent manner (***Ganapathy et al., 2008***). To characterize D-serine transport in both SMCT1 and SMCT2, we generated Flp-In T-REx 293 cells stably expressing human SMCT1 (FlpInTR-SMCT1) and SMCT2 (FlpInTR-SMCT2). Expression of SMCTs in FlpInTR-SMCT1 and FlpInTR-SMCT2 was verified by western blot using anti-FLAG antibody to detect FLAG-tagged SMCTs (***Figure 5—figure supplement 1A***). SMCT1 and SMCT2 expressions remained unchanged upon ASCT2 knockdown (***Figure 5—figure supplement 1A***). The function of SMCTs was confirmed by their canonical substrate [14C]nicotinate transport in the SMCT1- and SMCT2-stable cells (***Figure 5—figure supplement 1B and C***).

The transport functions of SMCTs are inhibited by non-steroidal anti-inflammatory drugs (e.g. ibuprofen and acetylsalicylate) (***Itagaki et al., 2006***; ***Gopal et al., 2007***). To verify the contribution of SMCTs to D-serine-reduced cell growth, we added ibuprofen to the growth assay. In contrast to mock, in which ibuprofen has no effect, the growth suppression by D-serine treatment was gradually attenuated by ibuprofen in both FlpInTR-SMCT1 and FlpInTR-SMCT2 cells, indicating that the cell growth suppression by D-serine treatment is the result of SMCT1 and SMCT2 functions (***Figure 5A***).

We characterized SMCTs mediating D-[3H]serine transport by using SMCT-stable cell lines. Both FlpInTR-SMCT1 and FlpInTR-SMCT2 cells transported D-[3H]serine over Mock (***Figure 5B***). In the cells

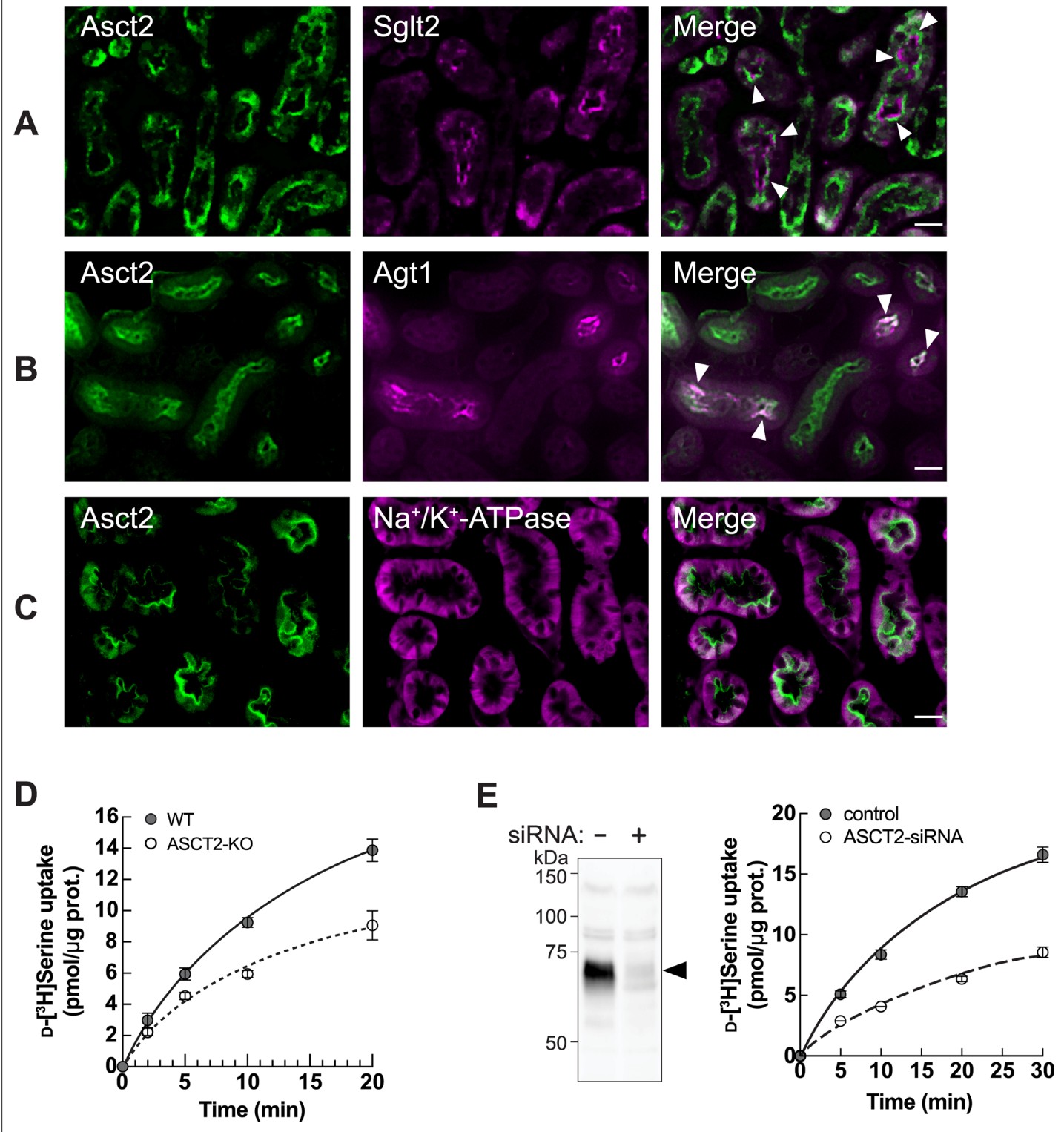

**Figure 3.** ASCT2 is one of D-serine transporters at the apical membrane of renal proximal tubular epithelia. (**A–C**) Localization of Asct2 in mouse kidney by immunofluorescence staining. Mouse kidney slides were co-stained with anti-Asct2(NT) antibody (Asct2; green) and protein markers for renal proximal tubule segments: anti-Sglt2 antibody (**A:** Sglt2, apical membrane marker of S1+S2 segments), anti-Agt1 antibody (**B:** Agt1, apical membrane marker of S3 segment), and anti-Na+/K+-ATPase antibody (**C:** Na+/K+-ATPase, basolateral membrane marker). Merge images are shown in the right panel. Arrowheads indicate co-localization of the proteins. Scale bar = 20 μm. (**D**) Time course of 100 μM D-[³H]serine transport in wild-type (WT) and *ASCT2* knockout (ASCT2-KO) HAP1 cells measured in PBS pH 7.4. Dot plot = mean ± SEM; n=3. (**E**) Left: Western blot using anti-ASCT2 antibody

*Figure 3 continued on next page*

Figure 3 continued

verified the suppression of ASCT2 in ASCT2-siRNA-transfected HEK293 cells. Right: Transport of 100 µM D-[³H]serine (in PBS pH 7.4) was measured in ASCT2 knockdown (ASCT2-siRNA) in comparison to the Mock cells (control). Dot plot = mean ± SEM; n=3.

The online version of this article includes the following source data and figure supplement(s) for figure 3:

**Source data 1.** The original unedited picture of western blot for *Figure 3E*.

**Source data 2.** The file containing *Figure 3E* and uncropped picture of western blot with indicated bands used in the figure.

**Figure supplement 1.** Characterization of ASCT2 as a D-serine transporter in HEK293 cells.

**Figure supplement 1—source data 1.** The original unedited picture of western blot:anti-Asct2(NT) for *Figure 3—figure supplement 1A*: left.

**Figure supplement 1—source data 2.** The original unedited picture of western blot:anti-Asct2(CT) for *Figure 3—figure supplement 1A*: right.

**Figure supplement 1—source data 3.** The file containing *Figure 3—figure supplement 1A* and uncropped picture of western blot with indicated bands used in the figure.

with ASCT2 knockdown, the background level was lower, thereby enhancing the D-[³H]serine transport contributed by both SMCT1 and SMCT2 (the net uptake after subtracted with background) (*Figure 5C*). The D-[³H]serine transport was inhibited by ibuprofen and acetylsalicylate, confirming the specific transport by SMCTs (*Figure 5D*). The excess of non-radioisotope-labeled D-serine also inhibited D-[³H]serine uptake in both SMCTs. Still, its inhibitory effect in SMCT2 was much less than in SMCT1, suggesting the lower affinity of D-serine in SMCT2 than in SMCT1 (*Figure 5D*).

## D-Serine transport properties and kinetics in SMCT1

Kinetics of D-[³H]serine transport in SMCT1 was measured in FlpInTR-SMCT1 cells in the presence of ASCT2 knockdown. SMCT1 transported D-[³H]serine in a concentration-dependent manner and the curve fitted to Michaelis-Menten kinetics with the apparent $K_m$ of 3.39±0.79 mM and $V_{max}$ of 18.23±1.73 pmol/µg protein/min (*Figure 5E*, *Figure 5—figure supplement 1D*).

The substrate selectivity of SMCT1 was investigated using synthetic biochemistry to avoid the interference of amino acid and carboxylate transports by endogenous transporters in cells. We established a cell-free assay system using proteoliposomes, in which purified proteins are reconstituted into liposomes and all substances in the system are controllable. We purified human SMCT1 by affinity column and reconstituted SMCT1-proteoliposome (SMCT1-PL) (*Figure 6A*). The function of SMCT1 in SMCT1-PL was verified by the transport of lactate, propionate, and nicotinate but not urate (negative control) (*Figure 6—figure supplement 1A*). The kinetics of [¹⁴C]nicotinate transport in SMCT1-PL revealed an apparent $K_m$ of 442±94 µM (*Figure 6—figure supplement 1B*), which is in a similar range to the $K_m$ previously described by electrophysiological experiments in *Xenopus* oocyte expression systems ($K_m$ 390 µM) (*Paroder et al., 2006*), confirming the functional properties of SMCT1 in our cell-free transport assay system.

The transport of D-[³H]serine in SMCT1-PL was Na⁺-dependent and reached the stationary phase at approximately 5 min (*Figure 6B*). The amount of D-serine transport was slightly lower than lactate and propionate but higher than L-serine and the transport was inhibited by ibuprofen, confirming SMCT1-mediated D-serine transport (*Figure 6C*). Amino acid selectivity revealed that SMCT1 recognized both L- and D-serine over other small amino acids (L- and D-alanine), acidic amino acids (L- and D-glutamate), and large neutral amino acids (L- and D-tyrosine) (*Figure 6D*). Therefore, we concluded that serine is the substrate of SMCT1 and the recognition is more stereoselective for the D- than the L-isomer.

## ASCT2 and SMCTs contribute to D-serine transport in renal proximal tubular epithelia

We next examined the contributions of ASCT2 and SMCTs to D-serine reabsorption in the kidney using an ex vivo transport assay. ASCT2 is an antiporter, influx one amino acid with efflux of another, while SMCTs are symporters (*Ganapathy et al., 2008*; *Scalise et al., 2018*). Therefore, we measured D-serine transport in the presence or absence of L-Gln preloading and with or without ibuprofen to distinguish the functions of mouse Asct2 and Smcts. The enantiomeric amino acid profile showed D-serine concentrations of 4–10 µM in the plasma samples (*Figure 1—figure supplement 1*) and 20–100 µM in the urine samples (*Figure 1—figure supplement 2*). Accordingly, we tested 10 µM D-[³H]serine transport in BBMVs derived from normal mice. In the non-preloading condition, D-[³H]serine uptake

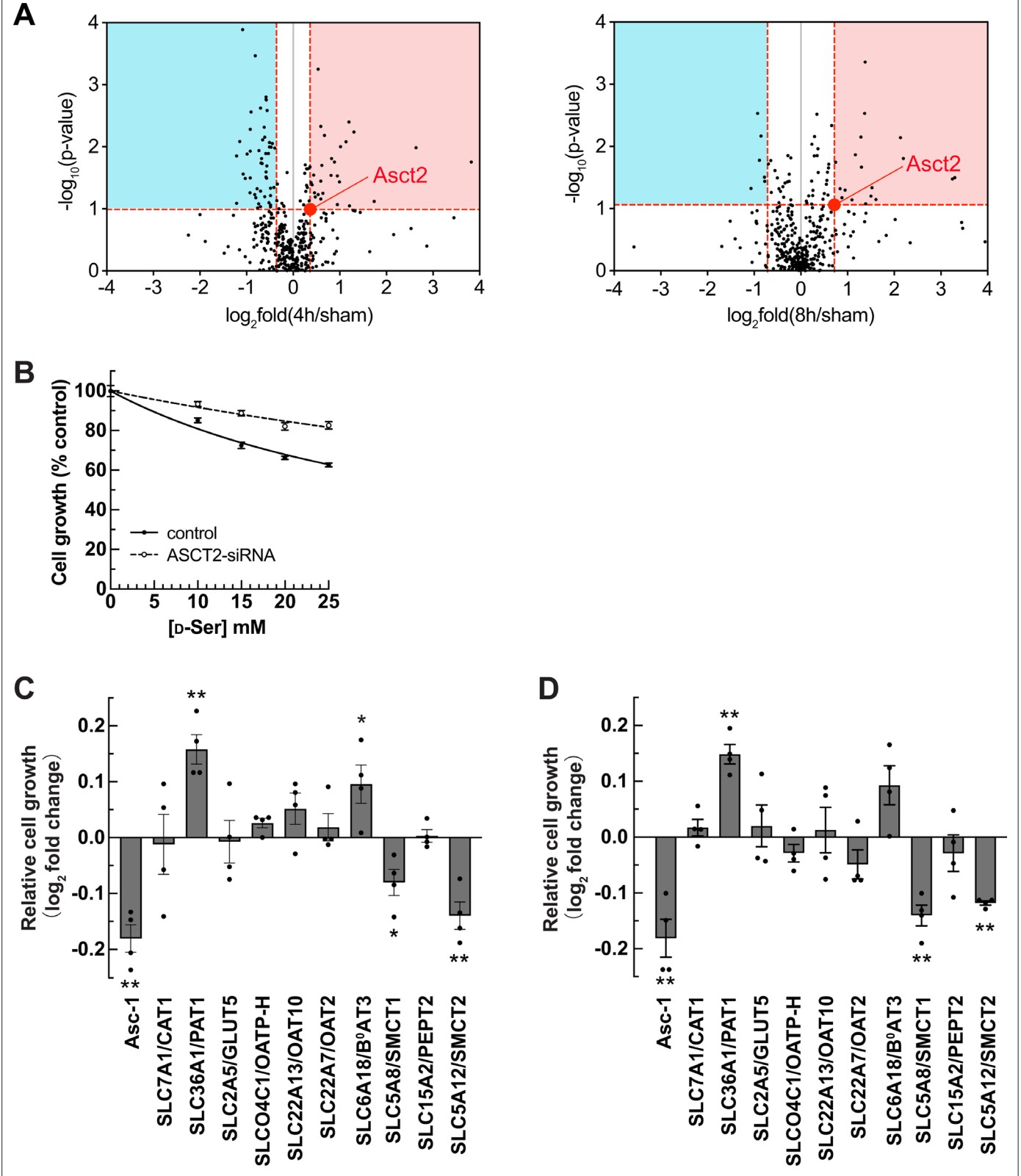

**Figure 4.** Identification of SMCT1 and SMCT2 as candidates of D-serine transporters. (**A**) Volcano plots of 398 membrane transport proteins identified from the brush border membrane vesicle (BBMV) proteome of the ischemia-reperfusion injury (IRI) model. The median of $\log_2$ fold of 4 hr IRI/sham (left) or 8 hr IRI/sham (right) were plotted against $-\log_{10}$ of p-value. Three proteins with $\log_2$ fold more than 4.0 (see values in **Supplementary file 1**) are omitted for a better view. The value of Asct2 (red dot) was set as a cutoff value to select D-serine transporter candidates (both increased [red area]

*Figure 4 continued on next page*

*Figure 4 continued*

and decreased [blue area] expressions). (**B**) Cell growth measurement (XTT assay) of ASCT2-siRNA or without siRNA (control) transfected HEK293 cells treated with D-serine. Data represent percent cell growth compared to the non-treated cells. The graphs were fitted to inhibition kinetics (dose-response – inhibition). Dot plot = mean ± SEM; n=5. (**C and D**) Candidates of D-serine transporters were screened by cell growth measurement. HEK293 cells were transfected with various cDNA clones, as indicated. After transfection, the cells were treated with either 15 mM (**C**) or 25 mM (**D**) D-serine for 2 days and cell growth was examined. The growth effect by D-serine treatment was normalized with that of no treatment and then calculated as $\log_2$ fold change of Mock at the same D-serine concentration. Asc-1 is used as the positive control. Bar graph = mean ± SEM; n=4; *p<0.05; **p<0.01. SMCT, sodium-coupled monocarboxylate transporter.

The online version of this article includes the following figure supplement(s) for figure 4:

**Figure supplement 1.** Expressions of Smct1, Smct2, Pat1, and $B^0$at3 in the volcano plots of membrane transport proteins.

in Na$^+$ dependence gradually but continuously increased and the uptake was remarkably inhibited by ibuprofen, suggesting that the transport was mainly attained from Smcts functions (***Figure 7A***). In L-Gln preloaded BBMVs, D-[$^3$H]serine uptake in Na$^+$ dependence arose quickly and reached the saturated point at 1–2 min (***Figure 7B***). The uptake at early time points (≤30 s) was ibuprofen-insensitive but became partly ibuprofen-sensitive from 1 min to the stationary phase (***Figure 7B***). These results indicate that the D-serine transport was mediated by both Asct2 and Smcts. At early time points, D-[$^3$H]serine was transported by Asct2, which occurred highly and rapidly. Meanwhile, Smcts functions (ibuprofen-sensitive) initiated later but had prolonged functions as seen in the non-preloading condition. Taken altogether, we suggested that, in the normal kidney where renal proximal tubular epithelial cells contain intracellular L-Gln, D-serine reabsorption is derived from combinational functions of both ibuprofen-sensitive (e.g. Smcts) and ibuprofen-insensitive (e.g. Asct2) transporters.

**Table 1.** Candidate transporters from proteomics of the brush border membrane vesicles (BBMVs) from the ischemia-reperfusion injury (IRI) model.
The list is ordered according to the fold change.

| Transporters | Accession | $\log_2$ fold 4 hr IRI/ sham | p-Value of 4 hr IRI/ sham | $\log_2$ fold 8 hr IRI/ sham | p-Value of 8 hr IRI/ sham | Peptides | Score Mascot | Abundance in sham* |
|---|---|---|---|---|---|---|---|---|
| *Increased* | | | | | | | | |
| Slc7a1/Cat1 | Q09143 | 0.8 | 0.12 | 1.3 | 0.01 | 1 | 131 | 2.6E+06 |
| Slc1a5/Asct2 | P51912 | 0.4 | 0.10 | 0.7 | 0.09 | 4 | 767 | 1.5E+07 |
| *Decreased* | | | | | | | | |
| Slc36a1/Pat1 | Q8K4D3 | −1.2 | 0.08 | −1.0 | 0.12 | 3 | 103 | 5.4E+06 |
| Slc2a5/Glut5 | Q9WV38 | −1.2 | 0.01 | −0.8 | 0.04 | 2 | 1,140 | 6.5E+07 |
| Slco4c1/ Oatp-m1 | Q8BGD4 | −1.1 | 0.00 | −0.9 | 0.00 | 9 | 4,011 | 1.5E+08 |
| Slc22a13/Oat10 | Q6A4L0 | −1.1 | 0.03 | −0.8 | 0.03 | 15 | 5,948 | 5.6E+08 |
| Slc22a7/Oat2 | Q91WU2 | −0.9 | 0.00 | −0.6 | 0.02 | 13 | 6,100 | 4.3E+08 |
| Slc6a18/$B^0$at3 | O88576 | −0.8 | 0.17 | −0.6 | 0.25 | 11 | 4,310 | 2.1E+08 |
| Slc5a8/Smct1 | Q8BYF6 | −0.8 | 0.05 | −0.7 | 0.02 | 25 | 26,044 | 2.9E+09 |
| Slc15a2/Pept2 | Q9ES07 | −0.7 | 0.01 | −1.1 | 0.05 | 9 | 2,379 | 4.6E+07 |
| Slc5a12/Smct2 | Q49B93 | −0.6 | 0.01 | −0.9 | 0.01 | 12 | 4,725 | 1.6E+08 |
| TMEM27/ Collectrin[†] | Q9ESG4 | −0.4 | 0.01 | −0.1 | 0.17 | 9 | 23,686 | 3.3E+09 |

*Median from n=3.

[†]Collectrin is a regulatory protein for $B^0$at3 function.

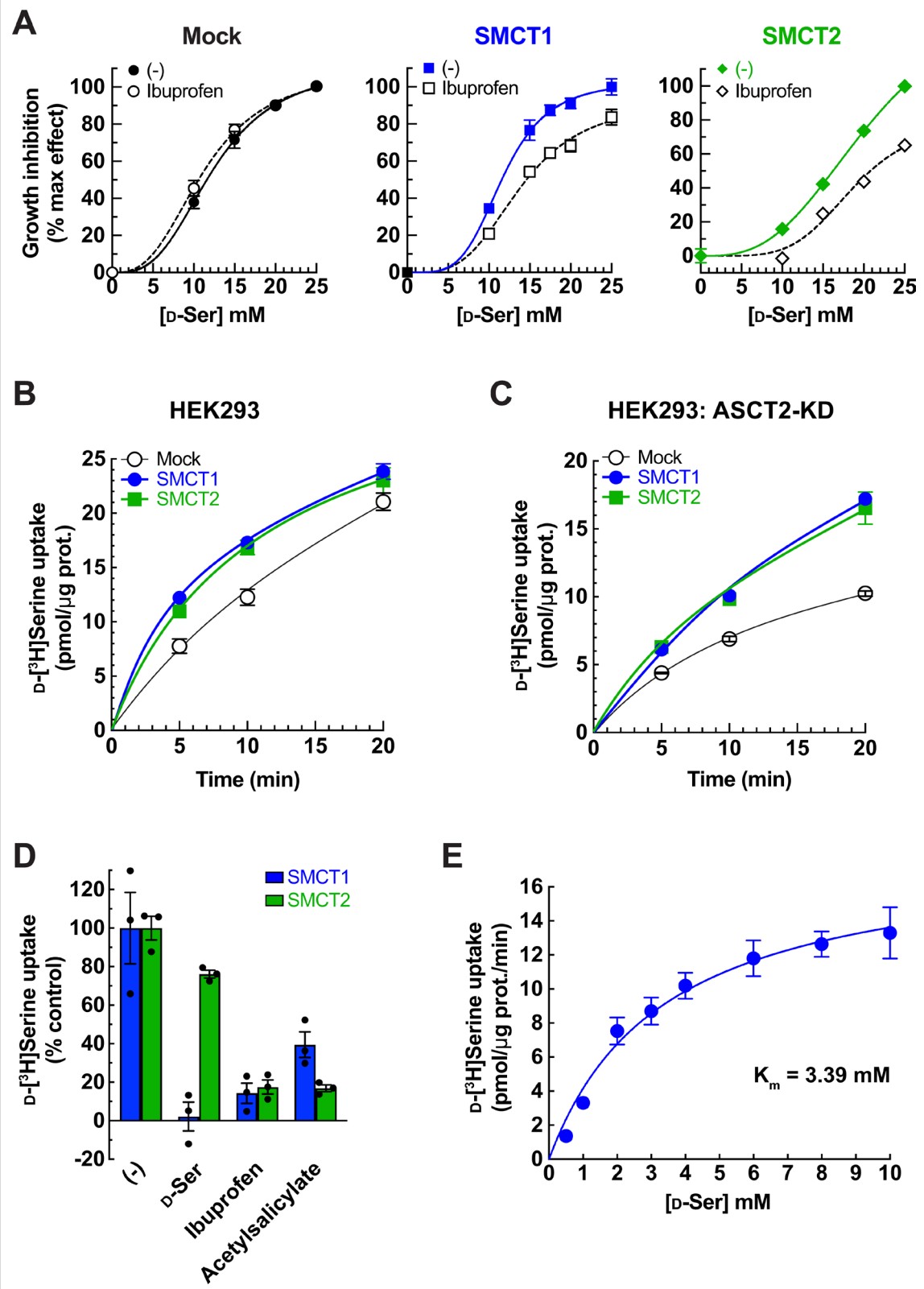

**Figure 5.** Characterization of SMCT1 and SMCT2 as D-serine transporters using SMCT1 and SMCT2 stably expressing cells. (**A**) Inhibition effect of ibuprofen on D-serine-induced cell growth. FlpInTR-Mock (Mock), FlpInTR-SMCT1 (SMCT1), or FlpInTR-SMCT2 (SMCT2) cells were treated with D-serine for 2 days in the presence or absence of 0.5 mM ibuprofen. Cell growth was measured by XTT assay. For comparison, the maximum growth inhibition by 25 mM D-serine treatment was set as 100% inhibition, and no D-serine treatment was set as 0% inhibition. The graphs were fitted to inhibition kinetics

*Figure 5 continued on next page*

Figure 5 continued

(dose-response – inhibition). Dot plot = mean ± SEM; n=5. (**B**) Time course of 100 µM D-[³H]serine uptake in FlpInTR-SMCT1 (SMCT1), FlpInTR-SMCT2 (SMCT2), and Mock cells. D-[³H]Serine transport was measured in PBS pH 7.4. Dot plot = mean ± SEM; n=4. (**C**) Time course of 100 µM D-[³H]serine uptake was measured similarly to (**B**), but the cells were subjected to ASCT2 knockdown (siRNA transfection) 2 days before the assay. Dot plot = mean ± SEM; n=4. (**D**) Transport of 20 µM D-[³H]serine by ASCT2-siRNA-transfected FlpInTR-SMCT1 or FlpInTR-SMCT2-stable cell lines were measured in the absence (-) or presence of 5 mM indicated inhibitors. The uptake was measured for 10 min in PBS pH 7.4. Graphs represented the uptake data subtracted from those of Mock cells. Bar graph = mean ± SEM; n=3. (**E**) Concentration dependence of D-[³H]serine transport in ASCT2-siRNA-transfected FlpInTR-SMCT1 cells. Uptake of D-[³H]serine (0.5–10 mM) was measured for 10 min in PBS pH 7.4. Raw data was shown in ***Figure 5—figure supplement 1D***. The uptake data in FlpInTR-SMCT1 were subtracted from those of Mock cells and fitted to Michaelis-Menten plot with the apparent $K_m$ of 3.39±0.79 mM and $V_{max}$ of 18.23±1.73 pmol/µg protein/min. Dot plot = mean ± SEM; n=3–4. SMCT, sodium-coupled monocarboxylate transporter.

The online version of this article includes the following source data and figure supplement(s) for figure 5:

**Figure supplement 1.** SMCT1 and SMCT2 functions in FlpInTR-SMCT1 and FlpInTR-SMCT2-stable cell lines.

**Figure supplement 1—source data 1.** The original unedited picture of western blot:anti-ASCT2 for ***Figure 5—figure supplement 1A***: top.

**Figure supplement 1—source data 2.** The original unedited picture of western blot:anti-FLAG for ***Figure 5—figure supplement 1A***: bottom.

**Figure supplement 1—source data 3.** The file containing ***Figure 5—figure supplement 1A*** and uncropped picture of western blot with indicated bands used in the figure.

---

The localization of Smct1 in the kidney was unclear. Gopal et al. reported the localization of Smct1 at S3 and Smct2 in all segments, while single-nucleus RNA sequencing (snRNA-seq) showed Smct1 at S2–S3 segments and Smct2 at S1 segment (***Figure 7—figure supplement 1***: controls; ***Gopal et al., 2007***; ***Kirita et al., 2020***). We generated an anti-Smct1 antibody and examined the localization of Smct1. The result showed that Smct1 was mainly localized at the S3 and slightly at the S1+S2 segments (***Figure 7—figure supplement 2***).

## D-Serine transport in renal proximal tubular epithelia of IRI model

Under IRI conditions, we observed low urinary but high plasma levels of D-serine, suggesting high D-serine reabsorption (***Figure 1—figure supplements 1A and 2A***). Our membrane proteomics revealed that Asct2 was increased while Smct1 and Smct2 were decreased in both 4 and 8 hr IRI (***Table 1***). We then examined the functional contributions of Asct2 and Smcts on the D-serine reabsorption during the pathology. Similar to those of the normal BBMVs (***Figure 7A and B***), D-[³H]serine transport in sham operation showed ibuprofen-sensitive in non-preloading and partly ibuprofen-sensitive in L-Gln preloading conditions (***Figure 7C***: sham). D-[³H]Serine influx reduced sharply in the BBMVs of 4 hr IRI model, and the transport was ibuprofen-insensitive in both with and without L-Gln preloading (***Figure 7C***: 4 hr IRI). Likely, this feature was due to the slight increase of Asct2 and the dramatic decrease of Smcts as observed in the proteome data (Asct2: 4 hr IRI/sham = 1.29 fold, Smct1: 4 hr IRI/sham = 0.59 fold; Smct2: 4 hr IRI/sham = 0.86 fold; ***Supplementary file 1***). We verified that BBMVs from all conditions contain functional proteins by measuring L-[³H]aspartate transport, which represented the function of Slc1a1/Eaac1 (***Kanai and Hediger, 1992***; ***Bailey et al., 2011***). The assay confirmed that BBMVs derived from 4 hr IRI model were not fully damaged, although the low influx was likely from the loss of protein expression during the injury (***Figure 2B***: amino acids, ***Figure 7—figure supplement 3***). In the BBMVs of 8 hr IRI model, D-[³H]serine transport was exceedingly high in L-Gln preloading and it was ibuprofen-insensitive (***Figure 7C***: L-Gln preloaded). These results could be explained by the continuing increase of Asct2 and decrease of Smcts as revealed by our proteomics (Asct2: 8 hr IRI/sham = 1.64 fold; Smct1: 8 hr IRI/sham = 0.62 fold; Smct2: 8 hr IRI/sham = 0.78 fold) (***Table 1***, ***Supplementary file 1***). The slight increase in D-[³H]serine transport in non-preloaded 8 hr IRI BBMVs may be due to an unknown transporter (***Figure 7C***: no preload). Our amino acid profiling showed that the amount of plasma and urinary L-Gln was not largely altered during 4 hr – 8 hr IRI (***Figures 1B and 2B***), suggesting that renal epithelial cells maintain intracellular L-Gln. Most likely, the high reabsorption of D-serine during IRI resulted from the increase of Asct2, despite the decreases of Smct1 and Smct2.

For comparison with our proteome data, we analyzed the mRNA expressions of Asct2, Smct1, and Smct2 in proximal tubule clusters at the early IRI from the open-source snRNA-seq dataset (***Kirita et al., 2020***). Consistent with our proteome data, Smct1 and Smct2 mRNA expressions were dramatically decreased, whereas Asct2 was increased since the early IRI (***Figure 7—figure supplement 1***). The correlation of proteomics and transcriptomics suggested that the protein alterations of Asct2 and

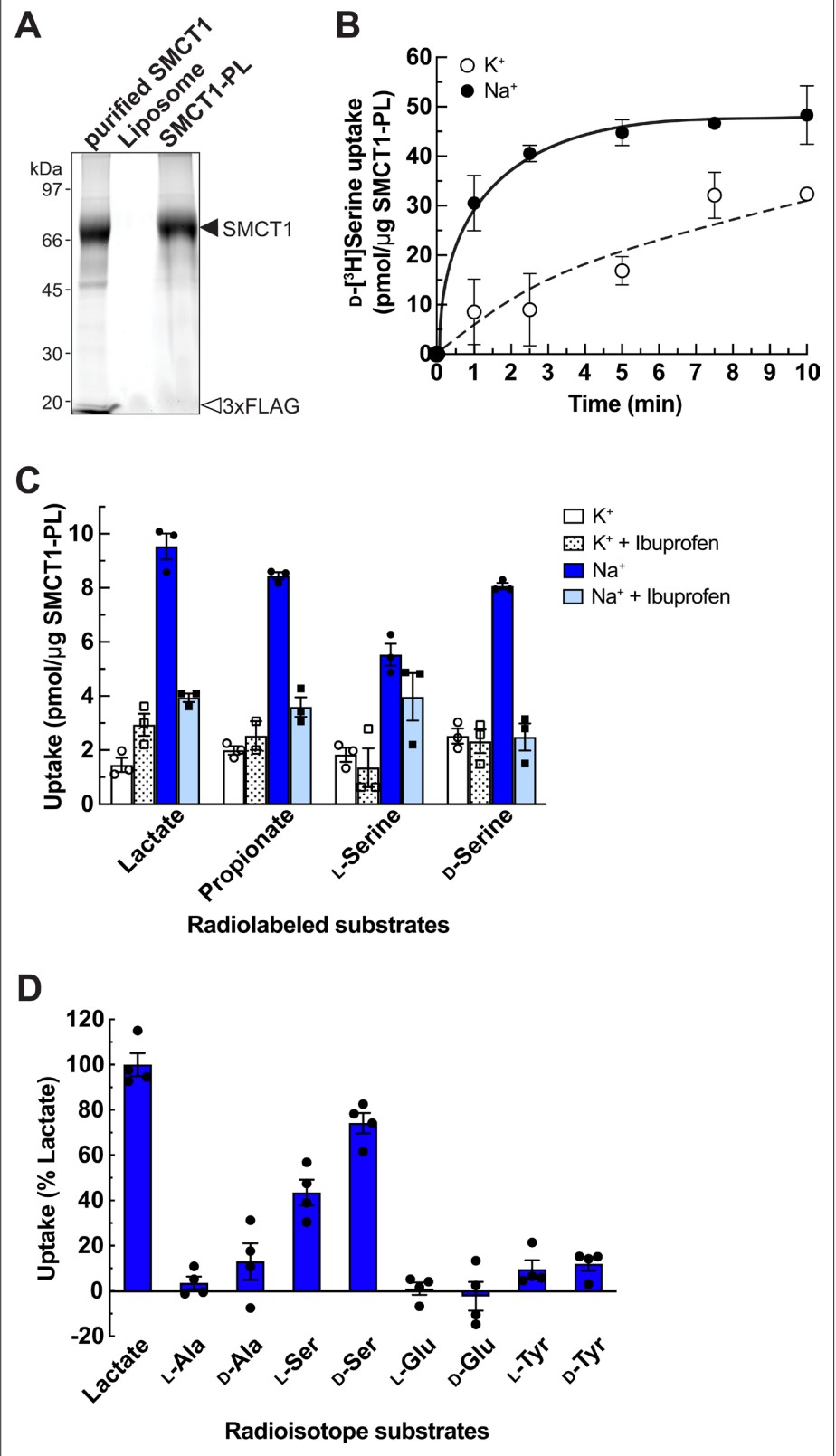

**Figure 6.** Characterization of SMCT1 as D-serine transporter using SMCT1-reconstituted proteoliposomes (SMCT1-PL). (**A**) Left: Stain-free SDS-PAGE gel shows SMCT1 (purified SMCT1) purified from pCMV14-SMCT1-transfected Expi293F cells, reconstituted empty liposomes (Liposome) and SMCT1-reconstituted proteoliposomes (SMCT1-PL). (**B**) Time course of D-[³H]serine transport in SMCT1-PL. Uptake of 200 µM D-[³H]serine was measured in

*Figure 6 continued on next page*

*Figure 6 continued*

Na$^+$-containing buffer (Na$^+$) compared to Na$^+$-free buffer (K$^+$). Dot plot = mean ± SEM; n=3. (**C**) Ibuprofen effect on the uptake of [$^3$H]lactate, [$^3$H]propionate, L-[$^3$H]serine, and D-[$^3$H]serine in SMCT1-PL. Uptakes of 50 µM radiolabeled substrates were measured for 5 min in Na$^+$-containing buffer (Na$^+$) or Na$^+$-free buffer (K$^+$) in the presence or absence of 1 mM ibuprofen. Bar graph = mean ± SEM; n=3. (**D**) Amino acid selectivity of SMCT1-PL. Transport of 50 µM radiolabeled amino acids was measured in SMCT1-PL for 5 min. The substrate uptake in Na$^+$-containing buffer was subtracted from those in Na$^+$-free buffer and calculated as % lactate uptake. Bar graph = mean ± SEM; n=4. SMCT, sodium-coupled monocarboxylate transporter.

The online version of this article includes the following source data and figure supplement(s) for figure 6:

**Source data 1.** The original unedited picture of SDS-PAGE gel for *Figure 6A*.

**Source data 2.** The file containing *Figure 6A* and uncropped picture of SDS-PAGE gel with indicated bands used in the figure.

**Figure supplement 1.** SMCT1 function in SMCT1-reconstituted proteoliposomes.

Smcts during the pathology emerged from transcriptional mechanisms rather than post-translational protein processing or protein degradation. When evaluating the extent of gene/protein alterations between the control and IRI conditions, we observed that the gene alterations of both Asct2 and Smcts, as revealed by snRNA-seq, are more pronounced than the protein alteration ratios obtained from proteomics. This discrepancy may stem from difficulty in the quantification method, especially for membrane transport proteins in label-free quantitative proteomics.

## Discussion

In this study, we presented an approach to investigate the transport systems for micronutrients and metabolites under physiological conditions and in multifactorial diseases. We selected AKI as a model study because AKI is known to target renal proximal epithelia and disrupt absorption/reabsorption systems that are the results of cooperative functions of multiple transporters. We utilized amino acid metabolomics to reveal a unique feature of the enantiomeric dynamics of serine in the AKI and analyzed membrane proteomes to obtain the D-serine transporter candidates. We then applied cell-based and cell-free transport assays to identify two D-serine transport systems: one was ASCT2 and the other consisted of unanticipated SMCTs (SMCT1 and SMCT2). Using ex vivo analysis of apical membrane-enriched renal proximal tubules, we showed that both transport systems contributed comparably to D-serine handling in the normal kidney, but ASCT2 became dominant in AKI, leading to the increase of D-serine in the blood. The alteration of two transport systems explained the dynamics of D-serine and suggested the transport mechanism behind the D-serine biomarker.

Several studies have reported the non-canonical substrates of membrane transport proteins, such as the recognition between amino acids, carboxylates, and amines among SLCs (*Metzner et al., 2005*; *Matsuo et al., 2008*; *Schweikhard and Ziegler, 2012*; *Wei et al., 2016*). Serine consists of a hydroxy-propionic acid and an amino group at the α-carbon. Possibly, the hydroxypropionic acid on serine is the main part to interact with SMCTs because hydroxypropionic acid shares similar moieties to lactic acid and propionic acid which are the high-affinity substrates of SMCTs. The apparent $K_m$ of D-serine transport in SMCT1 is 3.39 mM (*Figure 5E*), which is within the range of known SMCT1 substrates (0.07–6.5 mM) (*Ganapathy et al., 2008*), suggesting that SMCT1 accepts D-serine in the same manner as other monocarboxylate substrates. Inhibition of D-[$^3$H]serine transport by non-radioisotope-labeled D-serine in SMCT2 was less effective than in SMCT1 (*Figure 5D*). Together with the previous report that SMCT2 had lower inhibition affinities for nicotinate, lactate, and butyrate than SMCT1 (*Srinivas et al., 2005*), we suggest that SMCT2 recognizes D-serine with lower affinity than SMCT1.

Kinetics analysis of D-serine transport revealed the high affinity by ASCT2 ($K_m$ 167 µM) (*Foster et al., 2016*) and low affinity by SMCT1 ($K_m$ 3.39 mM; *Figure 5E*). In addition to transport affinity, the expression levels and co-localization of multiple transporters within the same cells are critical for elucidating the physiological roles of transporters or transport systems (*Sakaguchi et al., 2024*). In our proteome data, the chromatogram intensities of Smct1 ($2.9×10^9$ AU) and Smct2 ($1.6×10^8$ AU) were significantly higher than that of Asct2 ($1.5×10^7$ AU) in the control mice (*Table 1*: abundance in sham). While direct intensity comparisons between different proteins in mass spectrometry analyses are not precise, they can provide a general indication of relative protein amounts. This finding aligns with the snRNA-seq

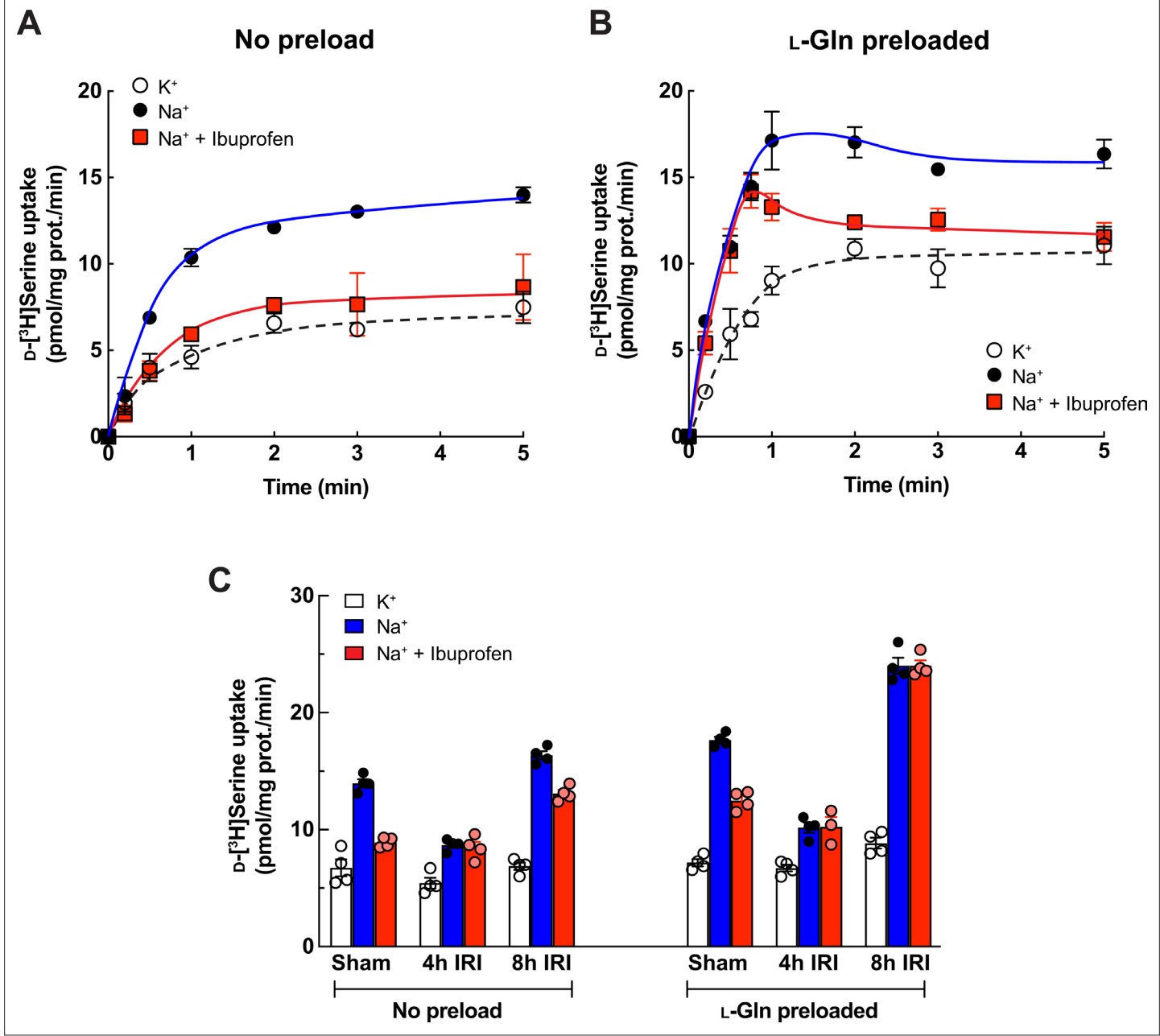

**Figure 7.** Characterization of D-serine transporters in brush border membrane vesicles (BBMVs) of normal mice and the ischemia-reperfusion injury (IRI) model. (**A**) Transport of 10 µM D-[³H]serine in renal BBMVs isolated from normal mice. The uptake was measured in Na⁺-free buffer (K⁺), Na⁺-containing buffer (Na⁺), or the presence of 1 mM ibuprofen (Na⁺+Ibuprofen). Dot plot = mean ± SEM; n=3. (**B**) Transport of 10 µM D-[³H]serine in renal BBMVs isolated from the normal mice was performed similarly to (**A**) but the BBMVs were preloaded with 4 mM L-Gln prior to the measurement. Dot plot = mean ± SEM; n=4. (**C**) Transport of 10 µM D-[³H]serine in renal BBMVs isolated from IRI model. Prior to uptake measurement, the BBMVs were preloaded with 4 mM L-Gln or buffer (no preload). The uptake was measured for 1 min in Na⁺-free buffer (K⁺), Na⁺-containing buffer (Na⁺), or the presence of 1 mM ibuprofen (Na⁺+Ibuprofen). Bar graph = mean ± SEM; n=3–4.

The online version of this article includes the following figure supplement(s) for figure 7:

**Figure supplement 1.** Expression of Smct1, Smct2, and Asct2 in renal proximal tubules of the ischemia-reperfusion injury (IRI) model by single-nucleus RNA sequencing (snRNA-seq).

**Figure supplement 2.** SMCT1 is mainly localized at the apical membrane of renal proximal tubular S3 segment.

**Figure supplement 3.** Functional L-aspartate transport in brush border membrane vesicles (BBMVs) from ischemia-reperfusion injury (IRI) model.

data, where Asct2 expression was found to be minimal, present in less than 10% of cell populations under both control and IRI conditions, suggesting that many cells do not express Asct2. Conversely, Smct1 and Smct2 show high and ubiquitous expression in control conditions, but their levels are markedly reduced in IRI conditions (*Figure 7—figure supplement 1*). Our ex vivo assays demonstrate that both ASCT2 and SMCTs mediate D-serine transport (*Figure 7B*). Consequently, Asct2 may contribute to D-serine reabsorption due to its high affinity. Meanwhile, Smcts, owing to their abundance particularly in cells lacking Asct2, likely play a significant role in D-serine reabsorption. Moreover, factors such as transport turnover rate ($K_{cat}$) and the presence of local canonical substrates are also vital in defining the overall contribution of D-serine transport systems.

Several pieces of evidence support SMCT1 at S3 segment and SMCT2 at S1 + S2 segments on renal D-serine handling. First, our enantiomeric amino acid profile from the IRI model, which agrees with the previous reports in CKD patients and AKI animal models, suggested the existence of a stereoselective transport system for serine (*Figure 1*; *Sasabe et al., 2014*; *Kimura et al., 2016*; *Sasabe and Suzuki, 2019*; *Hesaka et al., 2019*; *Kimura et al., 2020*). The system seems distinct from the known classical small amino acid transport systems and ASCT2. Second, previous studies indicated renal D-serine transport takes place at the proximal tubules by the distinct transport systems between S1+S2 and S3 segments. Both systems exhibited the characteristics of Na[+] dependency, electrogenicity, low affinity (mM range), and partial stereoselectivity (*Kragh-Hansen and Sheikh, 1984*; *Shimomura et al., 1988*; *Silbernagl et al., 1999*). These properties of the D-serine transport suit well to SMCTs and convince us of the contribution of SMCTs to renal D-serine transport in addition to ASCT2. Moreover, continuous administration of high doses of D-serine induces nephrotoxicity at the S3 segment of proximal tubules in normal animals (*Morehead et al., 1945*; *Silbernagl et al., 1999*; *Hasegawa et al., 2019*). It is most likely that the D-serine-induced proximal tubular damage is partly due to the absorption of D-serine by SMCTs, in particular by SMCT1 at the S3 segment. It is noted that SMCT2 is reported as a protein

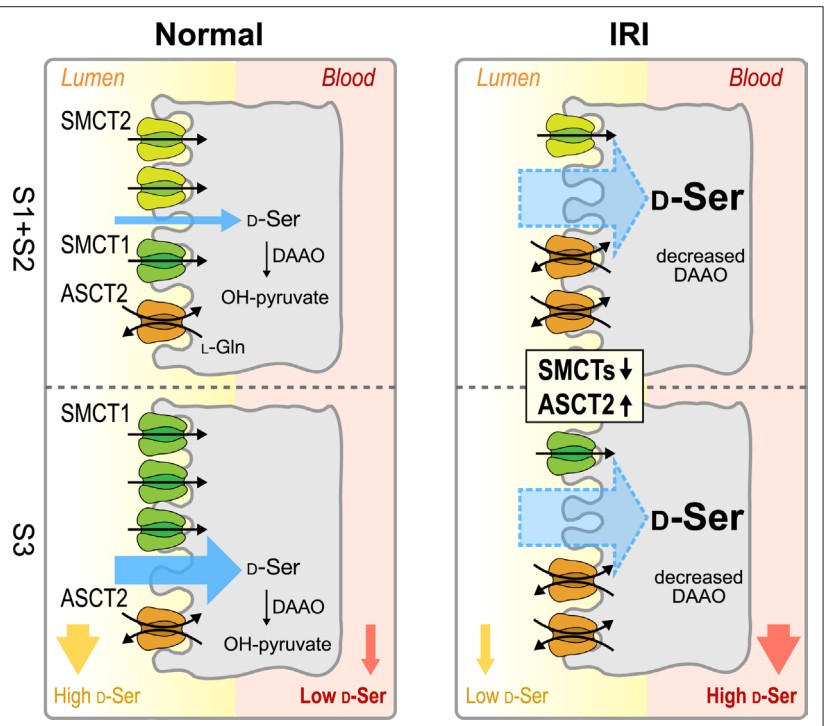

**Figure 8.** Proposed model of D-serine transport systems in renal proximal tubules. The model summarizes the contributions of ASCT2, SMCT1, and SMCT2 to D-serine transport in renal proximal tubules. Left: The physiological conditions (normal kidney). SMCT2 at S1 + S2 segments and SMCT1 at segment S3 have high expression levels, whereas the expression of ASCT2 is low. The higher D-serine affinity of SMCT1 compared to SMCT2 supports that the D-serine reabsorption tends to be exceeded at the S3 segment. Right: The pathological conditions (ischemia-reperfusion injury [IRI] model). Expressions of SMCTs decrease while that of ASCT2 increases. ASCT2 expresses ubiquitously and exhibits high D-serine affinity. Blue arrows anticipate total D-serine fluxes in each proximal tubular segment. SMCT, sodium-coupled monocarboxylate transporter.

indicator for kidney repair during the injury. Prolonged SMCT2 down-expression indicates the failed repair of proximal tubular cells in IRI (*Kirita et al., 2020*). The significance of SMCT2, in turn, supports the merit of D-serine usage in AKI and CKD diagnosis.

Combining all the results, we propose the model of D-serine transport systems in renal proximal tubules (*Figure 8*). In normal kidneys, the expression levels of SMCT2 at S1 + S2 segments and SMCT1 at S3 segment are high, whereas ASCT2 is low. Both SMCTs and ASCT2 are involved in D-serine reabsorption. Nonetheless, the net reabsorption levels appear to be relatively low due to the low affinities of SMCTs. It is likely that reabsorption at the S3 segment is dominated by SMCT1 function. The reabsorbed D-serine in proximal tubular cells is degraded into OH-pyruvate by D-amino acid oxidase (DAAO), thereby maintaining the low plasma D-serine but high urinary D-serine (*Figure 8*: normal). In pathological conditions such as AKI and CKD, the expression levels of SMCT1 and SMCT2 decrease dramatically while ASCT2 increases. The high affinity and ubiquitous expression of ASCT2 suggest the increased D-serine reabsorption in all proximal tubular segments. Together with the in-part reduction of DAAO activity during the pathology (*Sasabe et al., 2014*), the excess reabsorbed D-serine may lead to a high plasma D-serine level (*Figure 8*: IRI).

The enantiomeric profiles of serine revealed distinct plasma D-/L-serine ratios, with low ratios in the normal control but elevated ratios in IRI, despite the weak stereoselectivity of ASCT2 (*Figure 1B*). This observation suggested the differential renal handling of D-serine compared to L-serine. While we identified SMCTs as a D-serine transport system, it has been reported that L-serine reabsorption is mediated by $B^0AT3$ (*Singer et al., 2009*). We propose that the alterations in plasma and urinary D-/L-serine ratios are the combined outcomes of: (1) transport systems for L-serine and (2) transport systems for D-serine. In normal kidneys, the low plasma D-/L-serine ratios could result from the efficient reabsorption of L-serine by $B^0AT3$, coupled with the DAAO activity that degrades intracellular D-serine reabsorbed by SMCTs. In IRI conditions, our enantiomeric amino acid profiling revealed low plasma L-serine and high urinary L-serine (*Figure 1—figure supplements 1B and 2B*). Additionally, the proteomics analysis indicated a reduction in $B^0AT3$ levels (4 hr IRI/sham = 0.56 fold; 8 hr IRI/sham = 0.65 fold; *Supplementary file 1*). These observations suggest that the low L-serine reabsorption in IRI is a result of $B^0AT3$ reduction.

In proteomics, some transporters including ASCT2 were found to be increased in the AKI conditions (*Figure 2B*), yet the mechanism behind this change is still unclear. snRNA-seq study revealed that the cell clusters in all proximal tubular segments during early AKI were drastically different from the healthy kidney suggesting that the new distinct cell clusters in AKI were de novo synthesized for the repair process, as early AKI is a reversible condition (*Kirita et al., 2020*). ASCT2 was found to be only one type of small amino acid transporter which was increased during AKI conditions (*Figure 2B*). ASCT2 was reported to be highly expressed in proliferative cells and stem cells and played a key role in the regulation of mammalian target of rapamycin signaling pathway and glutamine-mediated metabolism (*Formisano and Van Winkle, 2016*; *Kandasamy et al., 2018*; *Scalise et al., 2018*). We postulate that ASCT2 is a nascent transporter to provide small amino acids and glutamine for cell growth, as a molecular mechanism in de novo cellular synthesis for the recovery processes.

Besides ASCT2 and SMCTs, other transporters may contribute to D-serine handling in the kidney. Influxes of D-[³H]serine in the Na⁺-free condition of the normal mouse BBMVs accounted for ~30% of total uptake (*Figure 7A and B*). These are the results of the experimental background of non-specific accumulation of radiotracers and probably also Na⁺-independent transporter(s). In BBMVs from 8 hr IRI model, D-[³H]serine uptake without preloading suggested the existence of Na⁺-dependent transporters during the injury. In the D-serine transporter screening, we found that PAT1 and $B^0AT3$ attenuated D-serine-mediated growth inhibition (*Figure 4C and D*). PAT1 is reported to transport D-serine in the transfected cell and oocyte models while the recognition of D-serine by $B^0AT3$ is unknown (*Boll et al., 2002*; *Chen et al., 2003*). PAT1, a low-affinity proton-coupled amino acid transporter ($K_m$ in mM range), has been found at both sub-apical membranes of the S1 segment and inside of the epithelia (The Human Protein Atlas: https://www.proteinatlas.org; updated on December 7, 2022) (*Sagné et al., 2001*; *Vanslambrouck et al., 2010*). PAT1 exhibits optimum function at pH 5–6 but very low activity at pH 7 (*Miyauchi et al., 2005*; *Bröer, 2008b*). Future research is required to address the significance of PAT1 on D-serine transport in the proximal tubule segments where pH regulation is known to be highly dynamic (*Boron, 2006*; *Nakanishi et al., 2012*; *Bouchard and Mehta, 2022*; *Imenez Silva and Mohebbi, 2022*). Despite the D-serine transport systems at the apical side, other

transport systems at the basolateral side are also necessary to complete the picture of the D-serine transport systems in renal proximal tubules.

Transport of a micromolecule is contributed by multiple transporters that orchestrate together to maintain homeostasis and to control transport compensation in an aberrant condition (*Bröer et al., 2016*; *Wiriyasermkul et al., 2020*; *Gauthier-Coles et al., 2021*; *Sakaguchi et al., 2024*). The molecular characteristics and physiological functions of transporters are largely dependent on their microenvironments, such as electrochemical gradients and the presence of other transporters for coordinating functions. In our proteomic analysis, a high population of membrane transport proteins was obtained in the dataset by applying urea treatment to the membrane preparation process (*Figure 2*, *Supplementary file 1*; *Kongpracha et al., 2022*). By integrating data focused on specific targets, such as transporters or amino acids, from the protein molecule level to the metabolome or proteome, we were able to reveal the subtle changes in the body and identify the causes of those small changes. This integrative and multi-hierarchical approach, emphasizing the coordinate functions of multiple transporters, holds the potential for investigating the dynamics of a broad range of micromolecules under diverse or intricate conditions within any tissues or organs.

# Materials and methods
## Materials, cell lines, animals, and graphical analysis

General chemicals and cell culture media were purchased from Wako Fujifilm and Nacalai tesque. Chemicals used in mass spectrometry were HPLC or MS grades. Expi293 Expression medium, Lipofectamine 3000, fluorescent-labeled secondary antibodies, and Tyramide SuperBoost kit were from Thermo. Amino acids, fetal bovine serum (FBS), anti-FLAG antibody, and anti-FLAG M2 column were from Sigma. Secondary antibodies with HRP conjugated were from Jackson ImmunoResearch. L-[$^3$H] Serine, D-[$^3$H]serine, L-[$^3$H]alanine, and D-[$^3$H]alanine were from Moravek. L-[$^3$H]Glutamic acid, D-[$^3$H] glutamic acid, L-[$^{14}$C]tyrosine, D-[$^{14}$C]tyrosine, DL-[$^3$H]lactic acid, [$^3$H]propionic acid, [$^{14}$C]nicotinic acid, [$^{14}$C]uric acid, and L-[$^3$H]aspartic acid were from American Radiolabeled Chemicals. Anti-SGLT2 (SC-393350) and anti-Na$^+$/K$^+$-ATPase (SC-21712) antibodies were obtained from Santa Cruz Biotechnology.

Flp-In T-Rex 293 cells and Expi293F cells were from Thermo. HEK293 was from ATCC. WT and *ASCT2* knockout HAP1 cells were from Horizon Discovery. The identities of all cell lines were authenticated using STR methods and verified for negative mycoplasma contamination by the manufacturers.

Eight-week-old C57BL/6J mice were purchased from Japan SLC and CLEA Japan. Frozen kidneys of the 8-week-old C57BL/6J mice were purchased from Sankyo Labo Service. All animal experiments complied with established guidelines and regulations under the ethical approval of The Institutional Animal Care and Use Committees of The Jikei University School of Medicine (Approval No. 2021-098), Nara Medical University, and Keio University, Japan.

Human genes/proteins were defined with all letters in uppercase while those of the mouse molecules were defined with the first letter in uppercase. Unless otherwise indicated, data shown in all figures are mean ± SEM of the representative data from three reproducible experiments. Statistical differences and p-values were determined using the unpaired Student's t test. Graphs, statistical significance, and kinetics were analyzed and plotted by GraphPad Prism 8.4.

## Plasmid construction

In this study, we used the cDNA of human *SLC5A8/SMCT1* clone 'NM_145913'. We generated the clone NM_145913 from the clone AK313788 (NBRC, NITE, Kisarazu, Japan). At first, *SMCT1* from AK313788 was subcloned into p3XFLAG-CMV14 (Sigma) via *Hind*III and *Bam*HI sites. The clone 'pCMV14-SMCT1' NM_145913 was subsequently generated by mutagenesis of p3XFLAG-CMV14-SMCT1_AK313788 at I193V, T201A, and I490M (variants between NM_145913 and AK313788) by HiFi DNA Assembly Cloning (NEB) and site-directed mutagenesis. Human *SLC5A12/SMCT2* cDNA (NM_178498; Sino Biological Inc) was amplified by PCR and cloned into p3XFLAG-CMV14 via *Kpn*I and *Bam*HI sites to generate 'pCMV14-SMCT2'. Human expression clones of *SLC7A10/Asc-1* (NM_019849), *SLC7A1/CAT1* (NM_003045), *SLC36A1/PAT1* (NM_078483), *SLC2A5/GLUT5* (NM_003039), *SLCO4C1/OATP-H* (NM_180991), *SLC22A13/OAT10* (NM_004256), *SLC22A7/OAT2* (NM_153320), *SLC6A18/B$^0$AT3* (NM_182632), *SLC15A2/PEPT2* (NM_021082), and *TMEM27/Collectrin* (NM_020665) were obtained from GenScript and RIKEN BRC through the National BioResource Project of the MEXT/AMED, Japan.

Mouse *Asct2* (NM_009201) TrueORF clone was obtained from OriGene. p3XFLAG-CMV14 empty vector was used for Mock production. pcDNA5-SMCT1 and pcDNA5-SMCT2 used for generation of SMCT1- and SMCT2-stable cell lines, respectively, were constructed by assembling the PCR products of SMCT1 or SMCT2 into pcDNA5/FRT/TO (Thermo) by HiFi DNA Assembly Cloning.

## Antibody production

Anti-mouse Smct1 antibody was custom-produced. Peptide antigen corresponding to the amino acid residues 596–611 of Smct1 and the corresponding anti-sera from the immunized rabbit was produced by Cosmo Bio. Anti-Smct1 antibody was purified using HiTrap Protein G HP (GE Healthcare) following the manufacturer's protocol. After purification, the antibody was dialyzed against PBS pH 7.4 and adjusted concentration to 1 mg/mL.

Anti-human ASCT2 and anti-mouse Asct2 antibodies were custom-produced. First, we generated pET47b(+)-GST vector by subcloning the *GST* from pET49b(+) template into pET47b(+) via *Not*I and *Xho*I restriction sites. Then, human ASCT2 antigen (amino acid residues 7–20) was amplified and cloned into pET47b(+)-GST to obtain GST-fused ASCT2 antigen. For the Asct2 antigen, both N-terminal (Asct2(NT), amino acid residues 1–38) and C-terminal (Asct2(CT), amino acid residues 521–553) fragments were fused with GST by cloning into pET47b(+)-GST and pET49b(+), respectively (Cosmo Bio). The GST-fusion antigens were expressed in *Escherichia coli* BL21(DE3) and purified by Gluta-thione Sepharose 4B (GE Healthcare) as described previously (*Nagamori et al., 2016a*). The antigens were used to immunize rabbits to obtain anti-sera (Cosmo Bio).

Anti-ASCT2 antibody was purified from the anti-sera using HiTrap Protein G HP (GE Healthcare) following the manufacturer's protocol. After purification, the antibody was dialyzed against PBS pH 7.4 and adjusted concentration to 1 mg/mL.

Anti-Asct2(NT) and anti-Asct2(CT) antibodies were purified by using two-step purifications. First, affinity columns of GST-fused Asct2(NT)-antigen and GST-fused Asct2(CT)-antigen were produced by conjugating the antigens with HiTrap NHS-activated HP (GE Healthcare). To purify anti-Asct2(NT) antibody, the anti-sera was subjected to the first purification by using the GST-fused Asct2(NT)-antigen column. The elution fraction was dialyzed and subsequently subjected to the second column of GST-fused Asct2(CT)-antigen column. The flowthrough fraction corresponding to the affinity-purified anti-Asct2(NT) antibody was obtained. Purification of anti-Asct2(CT) antibody was performed in the same way as Asct2(NT) antibody but used GST-fused Asct2(CT)-antigen column followed by GST-fused Asct2(NT)-antigen column.

## IRI model

C57BL/6J (CLEA Japan) male mice between 12 and 16 weeks of age underwent experimental proce-dures for the IRI model. Ischemia-reperfusion was performed as previously described (*Sasabe et al., 2014*). Before IRI induction, right kidney was removed. After 12 days, the mice were grouped by randomization. Ischemia was operated for 45 min by clamping the vessel under anesthesia. After that, the vessel clamp was removed, and the abdomen was closed. Sham-operated control mice were treated identically except for clamping. Urine and blood samples were collected at 4, 8, 20, and 40 hr after reperfusion. The mice were anesthetized with isoflurane and euthanized by perfusion with PBS pH 7.4. The kidney was removed and stored at –80°C until use.

## Measurement of amino acid enantiomers by 2D-HPLC

Plasma and urine samples collected from IRI mice were prepared as described with modifications (*Miyoshi et al., 2009*; *Hamase et al., 2010*). Briefly, 20-fold volumes of methanol were added to the samples and an aliquot was placed in a brown shading tube and used for NBD derivatization (1 µL of the plasma was used for the reaction). After drying the solution under reduced pressure, 20 µL of 200 mM sodium borate buffer (pH 8.0) and 5 µL of fluorescent-labeling reagent (40 mM 4-fluoro-7-ni tro-2,1,3-benzoxadiazole [NBD-F] in anhydrous acetonitrile [MeCN]) were added, and then heated at 60°C for 2 min. An aqueous 0.1% (vol/vol) TFA solution (375 µL) was added, and 20 µL of the reaction mixture was applied to 2D-HPLC.

Enantiomers of amino acids were quantified using the 2D-HPLC platform (*Miyoshi et al., 2009*; *Hamase et al., 2010*). The NBD derivatives of the amino acids were separated using a reversed-phase column (Singularity RP column, 1.0 mm i.d. × 50 mm; designed by Kyushu University and KAGAMI

INC) with the gradient elution using aqueous mobile phases containing MeCN and formic acid. To determine D- and L-amino acids, the fractions of amino acids were automatically collected using a multi-loop valve and transferred to the enantioselective column (Singularity CSP-001S, 1.5 mm i.d. × 75 mm; designed by Kyushu University and KAGAMI INC). Then, D- and L-amino acids were separated in the second dimension by the enantioselective column. The mobile phases are the mixed solutions of MeOH-MeCN containing formic acid, and the fluorescence detection of the NBD amino acids was carried out at 530 nm with excitation at 470 nm. Target peaks were quantified by scaling the standard peak shapes (*Hamase et al., 2018*).

## Isolation of BBMVs for mass spectrometry analysis

BBMVs were prepared by the calcium precipitation method (*Biber et al., 2007*; *Kongpracha et al., 2022*). Frozen kidneys were minced into fine powders by using a polytron-type homogenizer (Physcotron, Microtec) in the homogenization buffer containing 20 mM Tris-HCl, pH 7.6, 250 mM sucrose, 1 mM EDTA and cOmplete EDTA-free protease inhibitor cocktail (Roche). After low-speed centrifugation at 1000×g and 3000×g, the supernatant was collected and incubated with 11 mM CaCl$_2$ for 20 min on ice with mild shaking. The supernatant was ultracentrifuged at 463,000×g for 15 min at 4°C. The pellet was resuspended in the homogenization buffer and repeated the steps of CaCl$_2$ precipitation. Finally, the pellet of BBMVs was resuspended in 20 mM Tris-HCl pH 7.6, and 250 mM sucrose. Membrane proteins of BBMVs were enriched by the Urea Wash method (*Kongpracha et al., 2022*). The urea-washed BBMV samples were subjected to sample preparation for mass spectrometry.

## Cell culture, transfection, and generation of stable cell lines

HEK293 and Flp-In T-Rex 293 cells were cultured in DMEM supplemented with 10% (vol/vol) FBS, 100 units/mL penicillin G, and 100 µg/mL streptomycin (P/S), and routinely maintained at 37°C, 5% CO$_2$, and humidity. For transfection experiments, the cells were seeded in antibiotic-free media for 1 day prior to transfection to obtain approximately 40% confluence. DNA transient transfection and ASCT2-siRNA (ID#s12916; Ambion) transfection were performed by using Lipofectamine 3000 following the manufacturer's protocol. The ratio of DNA:P3000:Lipofectamine 3000 is 1.0 µg:2.0 µL:1.5 µL. The ratio of siRNA:Lipofectamine 3000 is 10 pmol:1 µL. The cells were further maintained in the same media for 2 days prior to the assays.

Flp-In T-REx 293 stably expressing SMCT1 (FlpInTR-SMCT1) and SMCT2 (FlpInTR-SMCT2) were generated by co-transfection of pOG44 and pcDNA5-SMCT1 or pcDNA5-SMCT2 and subsequently cultured in the media containing 5 mg/L blasticidin and 150 mg/L hygromycin B for positive clone selection. Mock cells were generated in the same way using the empty plasmid. Expressions of SMCT1 in FlpInTR-SMCT1 and SMCT2 in FlpInTR-SMCT2 were induced by adding 1 mg/L doxycycline hyclate (Dox; Tet-ON system) at 1 day after seeding. The cells were further cultured for 2 days prior to performing experiments.

WT and *ASCT2* knockout HAP1 cells (human near-haploid cell line; Horizon Discovery) were cultured in IMDM supplemented with 10% (vol/vol) FBS, P/S, and routinely maintained at 37°C, 5% CO$_2$, and humidity.

Expi293F cells were cultured in Expi293 Expression medium at 37°C, 8% CO$_2$, and humidity. To express SMCT1 transiently, the cells were transfected with pCMV14-SMCT1 using PEI MAX pH 6.9 (MW 40,000; Polysciences) and cultured for 2 days.

## Mass spectrometry and proteome data analysis

Proteomics of mouse BBMVs was analyzed as described (*Uetsuka et al., 2015*; *Kongpracha et al., 2022*). After preparing urea-washed BBMVs, the samples were fractionated into four fractions by SDB-XC StageTips and desalted by C18-StageTips. Mass spectrometry was performed using the Q Exactive (Thermo) coupled with nano-Advance UHPLC (Michrom Bioresources). The UHPLC apparatus was equipped with a trap column (L-column ODS, 0.3×5 mm, CERI) and a C18 packed tip column (0.1×125 mm; Nikkyo Technos). Raw data from four fractions were analyzed using Proteome Discoverer 2.2 (Thermo) and Mascot 2.6.2 (Matrix Science). Data from four fractions were combined and searched for identified proteins from the UniProt mouse database (released in March 2019). The maximum number of missed cleavages, precursor mass tolerance, and fragment mass tolerance were set to 3, 10 ppm, and 0.01 Da, respectively. The carbamidomethylation Cys was set as a fixed

modification. Oxidation of Met and deamidation of Asn and Gln were set as variable modifications. A filter (false discovery rate <1%) was applied to the resulting data. For each mouse sample, the analysis was conducted twice and the average was used. One dataset was composed of three samples from each operation condition (n=3). Statistical analyses were determined using Proteome Discoverer 2.2 (Thermo). Statistical differences and p-values were determined using the unpaired Student's t test. Data represented the median ± SEM (**Supplementary file 1**).

Mass spectrometry of HEK293 cells was analyzed from crude membrane fractions. Membrane fractions from HEK293 cells were prepared from 3-day cultured cells as described (**Nagamori et al., 2016b**). Membrane proteins were enriched by the Urea Wash method, and tryptic peptides were subject for analysis as described above.

Proteome of BBMVs from mouse kidneys after ischemia operation for 4 hr or 8 hr was normalized to that of sham operation. The identified proteins were then subjected to annotate the biological functions by IPA (QIAGEN). Molecules from the dataset that met the cutoff of the Ingenuity Knowledge Base were considered for the analysis. A right-tailed Fisher's exact test was used to calculate the p-value determining the probability (z-scores) of the biological function. Protein localization in kidney segments was evaluated based on related literature and The Human Protein Atlas (http://www.proteinatlas.org: updated December 7, 2022).

## Effect of D-serine on cell growth

HEK293 cells were seeded into 96-well plate at 10,000 cells/well. Transient transfection was performed 12 hr after seeding followed by L- or D-serine treatment at 12 hr after that. In the case of FlpInTR-stable cell lines, if needed, ASCT2 siRNA was transfected 12 hr after seeding. Dox was added 1 day after seeding followed by treatment with L- or D-serine (in the presence or absence of ibuprofen as indicated) 10 hr after adding Dox. The cells were further maintained for 2 days. Cell growth was examined by XTT assay. In one reaction, 50 µL of 1 mg/mL XTT (2,3-bis-(2-methoxy-4-nitro-5-sulf ophenyl)-2$H$-tetrazolium-5-carboxanilide) (Biotium) was mixed with 5 µL of 1.5 mg/mL phenazine methosulfate. The mixture was applied to the cells and incubated for 4 hr at 37°C in the cell culture incubator. Cell viability was evaluated by measuring the absorbance at 450 nm. Cell growth in serine treatment samples was compared to the control (without serine treatment). For transporter screening, the growth of the transfected cells at a specific D-serine concentration was compared to that of Mock after normalization with no treatment.

## Transport assay in cultured cells

Transport assay in cells was performed as described previously with some modifications (**Wiriyasermkul et al., 2012**). Briefly, for D-[³H]serine transport in HEK293 cells, the cells were seeded into poly-D-lysine-coated 24-well plates at 1.2×10⁵ cells/well and cultured for 3 days. Unless indicated elsewhere, uptake of 20 µM (100 Ci/mol) or 100 µM (10 Ci/mol) D-[³H]serine was measured in PBS pH 7.4 at 37°C at the indicated time points. After termination of the assay, the cells were lysed. An aliquot was subjected to measure protein concentration, and the remaining lysate was mixed with Optiphase HiSafe 3 (PerkinElmer). The radioactivity was monitored using a β-scintillation counter (LSC-8000, Hitachi). In the experiments which determine the importance of Na⁺, Na⁺-HBSS or Na⁺-free HBSS (choline-Cl substitution) were used instead of PBS.

Transport assay in FlpInTR-stable cell lines was performed in a similar way to HEK293 cells. After cell seeding for 1 day, SMCT1 and SMCT2 expression were induced by adding Dox for 2 days. For the ASCT2 knockdown experiment, ASCT2 siRNA was transfected 12 hr prior to Dox induction. The time course of 100 µM D-[³H]serine transport (10 Ci/mol) was measured 37°C for 5–20 min. Inhibition assay was performed by adding the inhibitors at the same time with D-[³H]serine substrate. Kinetics of D-[³H]serine transport were examined by the uptake of D-[³H]serine at the concentration of 0.5–8 mM (0.125–2 Ci/mol) for 10 min at 37°C.

Transport of D-[³H]serine in WT and *ASCT2* knockout HAP1 was performed in a similar way to HEK293 and FlpInTR-stable cells but without the process of transfection.

## SMCT1 purification, proteoliposome reconstitution, and transport assay

Human SMCT1 was purified from SMCT1-expressing Expi293F cells. Membrane fraction and purification processes were performed as previously described with small modifications (*Nagamori et al., 2016a*). First, cell pellets were resuspended in 20 mM Tris-HCl pH 7.4, 150 mM NaCl, 10% (vol/vol) glycerol, and protease inhibitor cocktail (Roche). The crude membrane fraction was derived from sonication and ultracentrifugation (sonication method). Membrane proteins were extracted from the crude membrane fraction with 2% (wt/vol) DDM and ultracentrifugation. SMCT1 was purified by anti-FLAG M2 affinity column. Unbound proteins were washed out by 20 mM Tris-HCl pH 7.4, 200 mM NaCl, 10% (vol/vol) glycerol, and 0.05% (wt/vol) DDM. Then SMCT1 was eluted by 3xFLAG peptide in the washing buffer. Purified SMCT1 was concentrated by Amicon Ultra Centrifugal Filters 30K (Millipore).

The reconstitution of proteoliposomes was performed as described with minor modifications (*Lee et al., 2019*). The purified SMCT1 was reconstituted in liposomes (made from 5:1 [wt/wt] of type II-S PC: brain total lipid) at a protein-to-lipid ratio of 1:100 [wt/wt] in 20 mM MOPS-Tris pH 7.0 and 100 mM KCl.

Transport assay in proteoliposomes was conducted by the rapid filtration method (*Lee et al., 2019*). Uptake reaction was conducted by dilution of 1 µg SMCT1 proteoliposomes in 100 µL uptake buffer (20 mM MOPS-Tris pH 7.0, 100 mM NaCl for $Na^+$-buffer or KCl for $Na^+$-free buffer, 1 mM $MgSO_4$, and 1 mM $CaCl_2$) containing radioisotope-labeled substrates. The reaction was incubated at 25°C for an indicated time. The radioisotope-labeled substrates were used as follows: 5 Ci/mol for [$^{14}$C]uric acid, L-[$^{14}$C]tyrosine, and D-[$^{14}$C]tyrosine; 10 Ci/mol for DL-[$^3$H]lactic acid, L-[$^3$H]alanine, D-[$^3$H]alanine, L-[$^3$H]serine, D-[$^3$H]serine, L-[$^3$H]glutamic acid, and D-[$^3$H]glutamic acid; and 20 Ci/mol for [$^3$H]propionic acid. In the inhibition assay, 1 mM ibuprofen was applied at the same time with the radioisotope-labeled substrates. Kinetics of nicotinate transport were measured for 3 min using 0.01–2 mM [$^{14}$C]nicotinate (0.2–50 Ci/mol) in $Na^+$-containing buffer, and then subtracted from the uptake in $Na^+$-free buffer.

## Transport assay in mouse BBMVs

For the experiments in healthy kidneys, the kidneys were taken from 8-week-old C57BL/6J mice (Japan SLC, CLEA Japan, or Sankyo Labo Service). For the experiments in IRI model, both left and right kidneys from 8-week-old male normal control (Japan SLC; Sankyo Labo Service) or IRI model (sham, 4 hr IRI, and 8 hr IRI) were taken out after PBS perfusion and frozen until use. After mincing and homogenizing the frozen kidneys in the buffer containing 20 mM Tris-HCl pH 7.6, 150 mM mannitol, 100 mM KCl, 1 mM EDTA, and protease inhibitor cocktail, the BBMVs were prepared as described in the above section but using magnesium instead of calcium for BBMV precipitation. The pellet of BBMVs was resuspended in the suspension buffer (10 mM Tris-HCl, pH 7.6, 100 mM mannitol, and 100 mM KCl). In the L-glutamine preloading experiment, the BBMVs were incubated with 4 mM L-glutamine (L-Gln) or only buffer (no preload) on ice for 3 hr. External buffers were then removed by centrifugation at 21,000×*g* for 20 min and the BBMVs were resuspended in the suspension buffer.

Transport assay was performed by rapid filtration. Prior to initiating the reaction, 5 µM valinomycin was added to the BBMV samples. Transport assay was examined by diluting 100 µg BBMVs in 100 µL of the uptake buffer (10 mM Tris-HCl pH 7.6, 150 mM NaCl for $Na^+$ condition or KCl for $Na^+$-free condition, 50 mM mannitol, and 5 µM valinomycin) containing radioisotope substrates, 10 µM D-[$^3$H]serine (100 Ci/mol) or 10 µM L-[$^3$H]aspartic acid (100 Ci/mol), as described in the figures. The reaction was incubated at 30°C at an indicated time, then terminated by the addition of ice-cold buffer containing 10 mM Tris-HCl pH 7.6 and 200 mM mannitol and filtered through 0.45 µm nitrocellulose filter (Millipore), followed by washing with the same buffer once. The membranes were soaked in Clear-sol I (Nacalai Tesque), and the radioactivity on the membrane was monitored. For the inhibition experiments, the tested inhibitors were added into the D-[$^3$H]serine substrate solution at the same time.

## Western blot analysis

Expressions of targeting proteins from membrane fractions were verified by western blot analysis as described (*Nagamori et al., 2016b*). Membrane fractions were dissolved in 1% wt/vol DDM prior to the addition of the SDS-PAGE sample buffer. Signals of chemiluminescence (Immobilon Forte Western HRP substrate; Millipore) were visualized by ChemiDoc MP Imaging system (Bio-Rad).

## Immunofluorescence staining of mouse kidneys

The 8-week-old male C57BL/6J mice (Japan SLC) were anesthetized and fixed by anterograde perfusion via the aorta with 4% wt/vol paraformaldehyde in 0.1 M sodium phosphate buffer pH 7.4. The kidneys were dissected, post-fixed in the same buffer for 2 days, and cryoprotected in 10%, 20%, and 30% wt/vol sucrose. Frozen kidney sections were cut at 7 μm thickness in a cryostat (Leica) and mounted on MAS-coated glass slides (Matsunami). The sections were placed in antigen retrieval buffer (10 mM citrate and 10 mM sodium citrate), autoclaved at 121°C for 5 min, and washed by TBS-T (Tris-buffered saline [TBS] with 0.1% vol/vol Tween 20). Immunostaining was done by serial incubation with each antibody as below.

For Asct2 and Sglt2 co-immunostaining, the samples were incubated with 3% hydrogen peroxide solution for 10 min, washed with TBS, and incubated in Blocking One Histo (Nacalai tesque) for 15 min. The samples were then incubated with mouse anti-SGLT2 antibody diluted in immunoreaction enhancer B solution (Can Get Signal immunostain, TOYOBO) overnight at 4°C. Signal was enhanced by Alexa Fluor 568 Tyramide SuperBoost (TSA) kit, goat anti-mouse IgG, following the manufacturer's instruction (Thermo). The antibodies were then stripped by citrate/acetate-based buffer, pH 6.0, containing 0.3% wt/vol SDS at 95°C for 10 min (*Buchwalow et al., 2018*), washed by TBS, and incubated with Blocking One Histo. Asct2 staining was done as described (*Nagamori et al., 2016a*). Briefly, the samples were incubated with rabbit anti-Asct2(NT) antibody diluted in immunoreaction enhancer A solution (Can Get Signal immunostain) overnight at 4°C. After washing with TBS-T, the specimens were incubated with Alexa Fluor 488-labeled donkey anti-rabbit IgG.

For Asct2 and Agt1 co-immunostaining, signals of both antibodies were enhanced by TSA kit. First, the specimens were incubated with rabbit anti-Agt1(G) antibody (*Nagamori et al., 2016a*) overnight at 4°C followed by Alexa Fluor 568 TSA kit with goat anti-rabbit. The antibodies were then stripped. The specimens were incubated with rabbit anti-Asct2(NT) overnight at 4°C and then repeated the steps of TSA kit using Alexa Fluor 488, goat anti-rabbit.

Staining of Asct2 and Na$^+$/K$^+$-ATPase was performed without TSA enhancement. After blocking by Blocking One Histo, the samples were incubated with rabbit anti-Asct2(NT) antibody diluted in immunoreaction enhancer A solution overnight. The samples were washed with TBS-T, incubated with Alexa Fluor488-labeled donkey anti-rabbit IgG, and washed again. Non-specific staining was blocked by Blocking One Histo and the specimens were then incubated with mouse anti-Na$^+$/K$^+$-ATPase antibody diluted in immunoreaction enhancer B solution overnight at 4°C. The specimens were washed with TBS-T and incubated with Alexa Fluor568-labeled goat anti-mouse IgG for 1 hr.

Smct1 and Sglt2 were co-stained. After blocking by Blocking One Histo, the samples were incubated with rabbit anti-Smct1 antibody and mouse anti-SGLT2 diluted in immunoreaction enhancer B solution overnight at 4°C. The specimens were washed with TBS-T and incubated with Alexa Fluor488-labeled donkey anti-rabbit IgG and Alexa Fluor568-labeled goat anti-mouse IgG for 1 hr.

Prior to the co-staining of Smct1 and Agt1, anti-Agt1 antibody was conjugated with Alexa Fluor 568 dye via succinimidyl ester reaction. The antibody was adjusted to pH 8.3 and mixed with Alexa Fluor 568 NHS Ester (Thermo) for 2 hr. The reaction was stopped by incubation with 1 M ethanolamine for 1 hr. The unconjugated dye was removed by size exclusion using Bio-Spin P-30 gel columns (Bio-Rad) equilibrated in PBS pH 7.4. For Smct1 and Agt1 co-staining, after the blocking step, the samples were first stained with rabbit anti-Smct1 antibody diluted in immunoreaction enhancer A solution. Signal was enhanced by Alexa Fluor 488 TSA kit with goat anti-rabbit IgG. Subsequently, the specimen was washed with TBS-T and non-specific staining was blocked by Blocking One Histo. The samples were then incubated with Alexa Fluor 568-conjugated anti-Smct1 antibody diluted in Blocking One Histo overnight at 4°C.

All specimens were washed with TBS-T and mounted with Fluoromount (Diagnostic Biosystems). Imaging was detected using a KEYENCE BZ-X710 microscope. Images were color-adjusted using ImageJ ver. 1.51 (NIH). Images in *Figure 3B* were subjected to deconvolution process in Fiji/ImageJ2 ver. 2.9 prior to color adjustment.

## Analysis of Smcts and Asct2 from the open-sourced dataset of snRNA-seq of the IRI model

The dataset of snRNA-seq used is created from the mice after IRI and published by *Kirita et al., 2020*. The snRNA-seq dataset (GSE139107), which has already processed by zUMIs and SoupX, was

downloaded from GEO, a public functional genomics data repository (*Barrett et al., 2013*). The dataset revealed percentage of mitochondrial UMI counts of each cell (calculated by *PercentageFeatureSet* function of Seurat) to be <0.01% in more than 94% cells in each cluster. From the 'GSE139107_MouseIRI.metadata', we annotated the expression of Smct1, Smct2, and Asct2 in proximal tubule segments 1–3 (S1, S2, and S3) at the early stages of IRI (4 and 12 hr) compared to the control mice. Downstream analyses including normalization, scaling, and visualization were performed by Seurat v3.2, an R package designed for quality control, analysis, and exploration of single-cell RNA-seq data (*Stuart et al., 2019*). After selecting the cells in S1, S2, and S3 from the datasets (control, 4 hr IRI, and 12 hr IRI), we merged the data of the cells in the same cluster. The data were then calculated as log-normalized values by *NormalizeData* function and scaled by *ScaleData* function of Seurat. Visualization of the expression data was performed using Seurat plug-in of R software and ggplot2 plug-in of R software (*Wickham, 2010*). Dot plots of gene expression in each cluster after log-transformation and scaling were created by *Dotplot* function of Seurat.

## Acknowledgements

We greatly appreciate Noriyoshi Isozumi for preliminary proteomic analysis, Yuki Mori, Junko Iwatani, and Yuika Shimo for experimental assistance, Hiroshi Imoto, Eiichi Negishi, Maiko Nakane, and Shoto Ishigo for optimization of enantiomeric amino acid analysis, Genro Kashino and Haruka Minowa for management of radioisotope facilities, Rikako Furuya for crucial discussion, Xuan Trang T Nguyen for her native English proofreading, and Sae Ochi and Tomokazu Matsuura for their tremendous supports. We are especially grateful to Yoshinori Moriyama for his critical reading and suggestion. This work is partly supported by JSPS KAKENHI (JP22K06150) to PW and (JP21H03365) to SN; research grant from Takeda Science Foundation to PW; research grants from Gout and Uric Acid Foundation, Nakatani Foundation, Shiseido Company, Ltd., AMED (JP21ek0310012), and AMED-CREST (JP21gm0810010) to SN; and unrestricted fund provided from Dr. Jin and Mrs. Keiko Hanyu (Jiseikai Kajigaya Clinic) to SN and PW.

## Additional information

### Competing interests

Masashi Mita: founder and CEO of KAGAMI Inc, a startup company working on chiral amino acids analysis and research for medical applications. Shushi Nagamori: A patent (WO/2021/132691) has been applied by KAGAMI Inc, Nara Medical University,and NIBIOHN with P.W., S.M., P.K., T.K., M.Mit., and S.N. as inventors based on this research. The authors declare no potential conflicts of interest. The other authors declare that no competing interests exist.

### Funding

| Funder | Grant reference number | Author |
| --- | --- | --- |
| Japan Society for the Promotion of Science | JP22K06150 | Pattama Wiriyasermkul |
| Takeda Science Foundation | | Pattama Wiriyasermkul |
| Japan Society for the Promotion of Science | JP21H03365 | Shushi Nagamori |
| Japan Agency for Medical Research and Development | JP21ek0310012 | Shushi Nagamori |
| Japan Agency for Medical Research and Development | JP21gm0810010 | Shushi Nagamori |
| Gout and Uric Acid Foundation | | Shushi Nagamori |

| Funder | Grant reference number | Author |
|---|---|---|
| Shiseido Group | | Shushi Nagamori |

The funders had no role in study design, data collection and interpretation, or the decision to submit the work for publication.

## Author contributions

Pattama Wiriyasermkul, Data curation, Formal analysis, Funding acquisition, Validation, Investigation, Visualization, Methodology, Writing – original draft, Writing – review and editing; Satomi Moriyama, Masataka Suzuki, Formal analysis, Investigation; Pornparn Kongpracha, Data curation, Formal analysis, Investigation; Nodoka Nakamae, Yoko Tanaka, Akina Matsuda, Investigation; Saki Takeshita, Data curation, Formal analysis; Masaki Miyasaka, Formal analysis, Validation; Kenji Hamase, Formal analysis, Investigation, Methodology; Tomonori Kimura, Resources, Methodology; Masashi Mita, Conceptualization, Funding acquisition; Jumpei Sasabe, Resources, Validation, Investigation, Methodology; Shushi Nagamori, Conceptualization, Resources, Data curation, Supervision, Funding acquisition, Validation, Methodology, Writing – original draft, Project administration, Writing – review and editing

## Author ORCIDs

Pattama Wiriyasermkul https://orcid.org/0000-0001-7068-7969
Shushi Nagamori https://orcid.org/0000-0003-0203-2754

## Ethics

All protocols for animal experiments were carried out following institutional guidelines under the approval of the Animal Care and Use Committees of The Jikei University School of Medicine, Nara Medical University, and Keio University, Japan (Approval No. 2021-098).

Reviewer #1 (Public Review): https://doi.org/10.7554/eLife.92615.3.sa1
Reviewer #2 (Public Review): https://doi.org/10.7554/eLife.92615.3.sa2
Reviewer #3 (Public Review): https://doi.org/10.7554/eLife.92615.3.sa3
Author response https://doi.org/10.7554/eLife.92615.3.sa4

# Additional files

## Supplementary files

- Supplementary file 1. Proteomics of brush border membrane vesicles (BBMVs) from the ischemia-reperfusion injury (IRI) model.
- Supplementary file 2. Annotation of membrane transport proteins from brush border membrane vesicle (BBMV) proteomics.
- Supplementary file 3. Proteomics of HEK293 membrane.
- MDAR checklist

## Data availability

All proteomics data have been deposited in Japan Proteome Standard Repository/Database: JPST000929 and JPST000931. Codes for snRNA sequencing reanalysis are available in GitHub repository at https://github.com/SN-PW/snRNA_seq (copy archived at *SN-lab SN-PW, 2023*).

The following datasets were generated:

| Author(s) | Year | Dataset title | Dataset URL | Database and Identifier |
|---|---|---|---|---|
| Nagamori S | 2021 | Proteome analysis of membranes from renal ischemia-reperfusion injury mouse models | https://repository.jpostdb.org/entry/JPST000929 | Japan Proteome Standard Repository/Database, JPST000929 |
| Nagamori S | 2023 | Proteome analysis of membranes from HEK293 cells | https://repository.jpostdb.org/entry/JPST000931 | Japan Proteome Standard Repository/Database, JPST000931 |

The following previously published dataset was used:

| Author(s) | Year | Dataset title | Dataset URL | Database and Identifier |
|---|---|---|---|---|
| Humphreys BD | 2020 | Cell profiling of acute kidney injury reveals conserved cellular responses to injury | https://www.ncbi.nlm.nih.gov/geo/query/acc.cgi?acc=GSE139107 | NCBI Gene Expression Omnibus, GSE139107 |

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
