## [Editor Report · eLife assessment]

This study shows **compelling** evidence that the less common D-serine stereoisomer is transported in the kidney by the neutral amino acid transporter ASCT2 and that it is a non-canonical substrate for sodium-coupled monocarboxylate transporter SMCTs. With a multi-hierarchical approach, this **important** study further shows that Ischemia-Reperfusion Injury in the kidney causes a specific increment in renal reabsorption carried out, in part, by ASCT2.

---

## [Referee Report · Reviewer #1 (Public Review)]

Most amino acids are stereoisomers in the L-enantiomer, but natural D-serine has also been detected in mammals and its levels shown to be connected to a number of different pathologies. Here, the authors convincingly show that D-serine is transported in the kidney by the neutral amino acid transporter ASCT2 and as a non-canonical substrate for the sodium-coupled monocarboxylate transporter SMCTs. Although both transport D-serine, this important study further shows in a mouse model for acute kidney injury that ASCT2 has the dominant role.

Strengths:

The paper combines proteomics, animal models, ex vivo transport analyses and in vitro transport assays using purified components. The exhaustive methods employed provide compelling evidence that both transporters can translocate D-serine in the kidney.

Weakness:

In the model for acute kidney injury the SMCTs proteins were not showing a significant change in expression levels and were rather analysed based on other, circumstantial evidence. Although its clear SMCTs can transport D-serine its physiological role is less obvious compared to ASCT2.

---

## [Referee Report · Reviewer #2 (Public Review)]

Summary:

The manuscript "A multi-hierarchical approach reveals D-1 serine as a hidden substrate of sodium-coupled monocarboxylate transporters" by Wiriyasermkul et al. is a resubmission of a manuscript, which focused first on the proteomic analysis of apical membrane isolated from mouse kidney with early ischemia- reperfusion injury (IRI), a well-known acute kidney injury (AKI) model. In a second part, the transport of D-serine by Asct2, Smct1, and Smct2 has been characterized in detail in different model systems, such as transfected cells and proteoliposomes.

Strengths:

A major problem with the first submission was the explanation of the link between the two parts of the manuscript: it was not very clear why the focus on Asct2, Smct1 and Smct2 was a consequence of the proteomic analysis. In the present version of the manuscript, the authors have focused on the expression of membrane transporters in the proteome analysis, thus making the reason for studying Asct2, Smct1 and Smct2 transporters more clear. In addition, the authors used 2D-HPLC to measure plasma and urinary enantiomers of 20 amino acids in plasma and urine samples from sham and ischaemia-reperfusion injury (IRI) mice. The results of this analysis demonstrated the value of D-serine as a potential marker of renal injury. These changes have greatly improved the manuscript and made it more convincing.

Weaknesses:

More than weakness I would speak of discussion points: I have a few suggestions that may help to make the paper more accessible to a general audience.

(1) In the Introduction, when the authors introduce the term "micromolecules", it would be beneficial to provide a precise definition or clarification of what they mean by this term. Adding a brief explanation may help the reader to better understand the context.

(2) In line 91, I suggest specifying that this is a renal IRI model.

(3) Lines 167-168 state that Asct2 is localised to the apical side of the renal proximal tubules. Is there any expression of Asct2 in other nephron segments?

(4) Lines 225-226: Have the authors expressed the candidate genes in HEK293 cells with ASCT2 knockdown?

(5) lines 254-255: why was D-serine transport enhanced by ASCT2 knockdown in FlpInTR-SMCT1 or 2 cells?

(6) line 265: The low affinity of SMCT1 for D-serine alone makes it an unlikely transporter for urinary D-serine.

(7) line 316: The authors state that there is a high tubular D-serine reabsorption in IRI and in line 424 that there is an inactivation of DAAO during the pathology. This suggests that there is a reabsorption of D-serine mediated by a transport system in the basolateral membrane domain of proximal tubular cells. Do the authors have any information about this transporter?

(8) in lines 462-463, the authors state: "It is suggested that PAT1 is less active at the apical membrane where the luminal pH is neutral". However, the pH of urine in the proximal tubules is normally acidic due to the high activity of NH3. I suggest rewording this sentence.

---

## [Referee Report · Reviewer #3 (Public Review)]

Summary:

The main objective of this work has been to delve into the mechanisms underlying the increment of D-serine in serum, as a marker of renal injury.

Strengths:

With a multi-hierarchical approach, the work shows that Ischemia reperfusion injury in kidney causes a specific increment in renal reabsorption of D-serine that, at least in part, is due to the increased expression of the apical transporter ASCT2. In the way, the authors revealed that SMCT1 also transports D-serine.

The manuscript also supports that increased expression of ASCT2, even together with the parallel decreased expression of SMCT1, in renal proximal tubules underlies the increased reabsorption of D-serine responsible of the increment of this enantiomer in serum in a murine model of ischemia reperfusion injury.

Weaknesses:

Remains to be clarified whether ASCT2 has substantial stereospecificity in favor of D- versus L-serine to sustain a ~10-fold decreased in the ratio D-serine/L-serine in the urine of mouse under ischemia reperfusion injury (IRI).

It is not clear how the increment in the expression of ASCT2, in parallel with the decreased expression of SMCT1, results in increased renal reabsorption of D-serine in IRI.

I am satisfied with the changes the authors have introduced in the text of the revised version of their manuscript.

---

## [Author Response]

The following is the authors’ response to the original reviews.

**eLife assessment**
This important study convincingly shows that the less common D-serine stereoisomer is transported in the kidney by the neutral amino acid transporter ASCT2 and that it is a noncanonical substrate for sodium-coupled monocarboxylate transporter SMCTs. With a multihierarchical approach, this important study further shows that Ischemia-Reperfusion Injury in the kidney causes a specific increment in renal reabsorption carried out, in part, by ASCT2.
**Public Reviews:**

**Reviewer #1 (Public Review):**
Most amino acids are stereoisomers in the L-enantiomer, but natural D-serine has also been detected in mammals and its levels shown to be connected to a number of different pathologies. Here, the authors convincingly show that D-serine is transported in the kidney by the neutral amino acid transporter ASCT2 and as a non-canonical substrate for the sodium-coupled monocarboxylate transporter SMCTs. Although both transport D-serine, this important study further shows in a mouse model for acute kidney injury that ASCT2 has the dominant role.Strengths:The paper combines proteomics, animal models, ex vivo transport analyses, and in vitro transport assays using purified components. The exhaustive methods employed provide compelling evidence that both transporters can translocate D-serine in the kidney.Weakness:In the model for acute kidney injury, the SMCTs proteins were not showing a significant change in expression levels and were rather analysed based on other, circumstantial evidence. Although its clear SMCTs can transport D-serine its physiological role is less obvious compared to ASCT2.

We greatly value the reviewer's efforts and feedback in reviewing our manuscript. We acknowledge the reviewer's observation that the changes indicated by our proteomic results are not markedly pronounced. To reinforce our findings, we have incorporated an analysis of gene alterations at the single-cell level (snRNA-seq) from the publicly accessible IRI mouse model data (Figure supplement 7). The snRNA-seq data align with our proteomic data in terms of the general trend of gene/protein alterations, but reveal more substantial changes in both ASCT2 and SMCTs. These discrepancies might stem from the different quantification methods used, suggesting a possible underestimation in our label-free proteomic quantification. The differences we see between the functional changes in transporters and their quantification in proteomics can be explained by the unique challenges posed by membrane proteins. Post-translational modifications and the complex nature of multiple transmembrane domains often impact the accurate measurement of these proteins in proteomic studies. This complexity can lead to a mismatch between the actual functional changes occurring in the transporters and their perceived abundance or alterations as detected by proteomic methods (Figure 4A) (Schey KL et al. Biochemistry 2015, doi: 10.1021/bi301604j). However, this label-free quantitative proteomics approach is well-suited for our study, given its screening efficiency, compatibility with animal models, and the absence of a labeling requirement. We may consider incorporating alternative quantitative proteomic methods in future for a more thorough comparison. We have included these considerations in lines 351-356 of the revised manuscript.

Manuscript lines 351-356

“When evaluating the extent of gene/protein alterations between the control and IRI conditions, we observed that the gene alterations of both Asct2 and Smcts, as revealed by snRNAsequencing, are more pronounced than the protein alteration ratios obtained from proteomics. This discrepancy may stem from difficulty in the quantification method, especially for membrane transport proteins in label-free quantitative proteomics.”

Regarding the roles of ASCT2 and SMCTs in renal D-serine transport, snRNA-seq showed that ASCT2 expression in the controls is less than 10% of the cell population. We suggest that ASCT2 contributes to D-serine reabsorption because of its high affinity and SMCTs (SMCT1 and SMCT2) would play a role in D-serine reabsorption in the cells without ASCT2 expression. In addition, we included other factors (the turnover rate and the presence of local canonical substrates) that may determine the capability of D-serine reabsorption. We have included this suggestion in the Discussion lines 386-404.

Manuscript lines 386-404

“Kinetics analysis of D-serine transport revealed the high affinity by ASCT2 (Km 167 µM) (Foster et al., 2016) and low affinity by SMCT1 (Km 3.39 mM; Figure 5E). In addition to transport affinity, the expression levels and co-localization of multiple transporters within the same cells are critical for elucidating the physiological roles of transporters or transport systems (Sakaguchi et al., 2024). In our proteome data, the chromatogram intensities of Smct1 (2.9 x 109 AU) and Smct2 (1.6 x 108 AU) were significantly higher than that of Asct2 (1.5 x 107 AU) in control mice (Table 1: abundance in Sham). While direct intensity comparisons between different proteins in mass spectrometry analyses are not precise, they can provide a general indication of relative protein amounts. This finding aligns with the snRNA-seq data, where Asct2 expression was found to be minimal, present in less than 10% of cell populations under both control and IRI conditions, suggesting that many cells do not express Asct2. Conversely, Smct1 and Smct2 show high and ubiquitous expression in control conditions, but their levels are markedly reduced in IRI conditions (Figure supplement 7). Our ex vivo assays demonstrate that both ASCT2 and SMCTs mediate D-serine transport (Figure 7B). Consequently, Asct2 may contribute to D-serine reabsorption due to its high affinity, whereas Smcts, owing to their abundance, particularly in cells lacking Asct2, likely play a significant role in D-serine reabsorption. Moreover, factors such as transport turnover rate (Kcat) and the presence of local canonical substrates are also vital in defining the overall contribution of Dserine transport systems.”

**Reviewer #2 (Public Review):**
Summary:The manuscript "A multi-hierarchical approach reveals D-1 serine as a hidden substrate of sodium-coupled monocarboxylate transporters" by Wiriyasermkul et al. is a resubmission of a manuscript, which focused first on the proteomic analysis of apical membrane isolated from mouse kidney with early Ischemia-Reperfusion Injury (IRI), a well-known acute kidney injury (AKI) model. In the second part, the transport of D-serine by Asct2, Smct1, and Smct2 has been characterized in detail in different model systems, such as transfected cells and proteoliposomes.Strengths:A major problem with the first submission was the explanation of the link between the two parts of the manuscript: it was not very clear why the focus on Asct2, Smct1, and Smct2 was a consequence of the proteomic analysis. In the present version of the manuscript, the authors have focused on the expression of membrane transporters in the proteome analysis, thus making the reason for studying Asct2, Smct1, and Smct2 transporters more clear. In addition, the authors used 2D-HPLC to measure plasma and urinary enantiomers of 20 amino acids in plasma and urine samples from sham and Ischemia-Reperfusion Injury (IRI) mice. The results of this analysis demonstrated the value of D-serine as a potential marker of renal injury. These changes have greatly improved the manuscript and made it more convincing.

We deeply appreciate the reviewer’s comments on the manuscript. We have responded to the recommendations one by one in the later section.

**Reviewer #3 (Public Review):**
Summary:The main objective of this work has been to delve into the mechanisms underlying the increment of D-serine in serum, as a marker of renal injury.Strengths:With a multi-hierarchical approach, the work shows that Ischemia-Reperfusion Injury in the kidney causes a specific increment in renal reabsorption of D-serine that, at least in part, is due to the increased expression of the apical transporter ASCT2. In this way, the authors revealed that SMCT1 also transports D-serine.The experimental approach and the identification of D-serine as a new substrate for SMCT1 merit publication in Elife.The manuscript also supports that increased expression of ASCT2, even together with the parallel decreased expression of SMCT1, in renal proximal tubules underlies the increased reabsorption of D-serine responsible for the increment of this enantiomer in serum in a murine model of Ischemia-Reperfusion Injury.Weaknesses:Remains to be clarified whether ASCT2 has substantial stereospecificity in favor of D- versus L-serine to sustain a ~10-fold decrease in the ratio D-serine/L-serine in the urine of mice under Ischemia-Reperfusion Injury (IRI).It is not clear how the increment in the expression of ASCT2, in parallel with the decreased expression of SMCT1, results in increased renal reabsorption of D-serine in IRI.

We thoughtfully appreciate the reviewer’s comment on the manuscript. Considering the alteration of D-/L-serine ratios, there are several factors including protein expression levels at both apical and basolateral sides, properties of the transporters (e.g. transport affinities, substrate stereoselectivities), and the expression of DAAO (D-amino acid oxidase) which selectively degrades D-amino acids. Moreover, the mechanism becomes more complicated when the transport systems of L- and D-enantiomers are different and have distinct stereoselectivities as in the case of serine. Future studies are required to complete the mechanism. However, we would like to explore the mechanism based on the current knowledge.

From this study, we identified ASCT2 and SMCTs (SMCT1 and SMCT2) as D-serine transport systems. We showed that SMCT1 prefers D-serine. Although we did not analyze ASCT2 stereoselectivity, based on the previous studies, ASCT2 recognizes both D- and Lserine with high affinities and slightly prefers L-enantiomer (Km of 18.4 µM) for L-serine in oocyte expression system (Utsunomiya-Tate et al. J Biol Chem 1996) and 167 µM for Dserine in oocyte expression system (Foster et al. Plos ONE 2016), and the IC50 of 0.7 mM for L-serine and 4.9 mM for D-serine (in HEK293 expression systems, Foster et al. PLOS ONE 2016). The proteomics showed an increase of ASCT2 (1.6-fold increase) and a decrease of SMCTs (1.7-fold decrease in SMCT1, and 1.3-fold decrease in SMCT2) in IRI conditions. The table below summarizes D-serine transport by ASCT2 and SMCTs.

In the case of L-serine, ASCT2 and B0ATs (in particular B0AT3) have been revealed as L-serine transport systems in the kidneys (Bröer et al. Physiol Rev 2008; Singer et al. J Biol Chem 2009). Proteomics showed that B0ATs have higher expression levels than ASCT2 supporting the idea that B0ATs are the main L-serine transport system (Table S1: Abundance of B0AT1 = 1.34E+09, B0AT3 = 2.13E+08, ASCT2 = 1.46E+07). In IRI conditions, B0AT3 decreased 1.8 fold and B0AT1 decreased 1.1 fold. From these results, we included the contribution of B0ATs in L-serine transport in Author response table 1.

**Author response table 1. sa4table1:** 

Transport systems	Normal condition		IRI	
	L-serine	D-serine	L-serine	D-serine
SMCTs	-	+	-	-
ASCT2	++	+	+++	++
B^(0)ATs	++++++	(-?)	+++	(-?)
DAAO		Degrade D-serine		No additionalrole

Taken together, we suggest that high ratios of D-/L-serine in IRI conditions are a combinational result of (1) increase of D-serine reabsorption by ASCT2 enhancement and SMCTs reduction and (2) decrease of L-serine reabsorption by B0ATs. We have included this suggestion in the Discussion lines 438-451.

Manuscript lines 438-451

“The enantiomeric profiles of serine revealed distinct plasma D/L-serine ratio, with low rations in the normal control but elevated ratios in IRI, despite the weak stereoselectivity of ASCT2 (Figure 1B). This observation suggested differential renal handling of D-serine compared to L-serine. While we identified SMCTs as a D-serine transport system, it has been reported that L-serine reabsorption is mediated by B0AT3 (Singer et al., 2009). We propose that the alterations in plasma and urinary D/L-serine ratios are the combined outcomes of: (1) transport systems for L-serine, and (2) transport systems for D-serine. In normal kidneys, the low plasma D/L-serine ratios could result from the efficient reabsorption of L-serine by B0AT3, coupled with the DAAO activity that degrades intracellular D-serine reabsorbed by SMCTs. In IRI conditions, our enantiomeric amino acid profiling revealed low plasma L-serine and high urinary L-serine (Figure supplements 1B, 2B). Additionally, the proteomic analysis indicated a reduction in B0AT3 levels (4h IRI/sham = 0.56 fold; 8h IRI/sham = 0.65 fold; Table S1). These observations suggest that the low L-serine reabsorption in IRI is a result of B0AT3 reduction.”

**Recommendations for the authors:**

**Reviewer #1 (Recommendations For The Authors):**
This is a thorough study that was reviewed previously under the old system. I think the authors have strengthened their findings and have no further suggestions.

We appreciate reviewer 1 for his/her effort and comments, which greatly contributed to improving this manuscript.

**Reviewer #2 (Recommendations For The Authors):**
The experiments seem to me to have been well performed and the data are readily available.Weaknesses:More than weakness I would speak of discussion points: I have a few suggestions that may help to make the paper more accessible to a general audience.(1) In the Introduction, when the authors introduce the term "micromolecules", it would be beneficial to provide a precise definition or clarification of what they mean by this term. Adding a brief explanation may help the reader to better understand the context.

Following the reviewer’s comment, we have included the explanation of the micromolecule and membrane transport proteins in lines 41-43.

Manuscript lines 41-43

“Membrane transport proteins function to transport micromolecules such as nutrients, ions, and metabolites across membranes, thereby playing a pivotal role in the regulation of micromolecular homeostasis.”

(2) In line 91, I suggest specifying that this is a renal IRI model.

Following the reviewer’s comment, we have added the information that it is a renal IRI model of AKI (lines 90-92).

Manuscript lines 90-92

“We applied 2D-HPLC to quantify the plasma and urinary enantiomers of 20 amino acids of renal ischemia-reperfusion injury (IRI) mice, a model of AKI and AKI-to-CKD transition (Sasabe et al., 2014; Fu et al., 2018).”

(3) Lines 167-168 state that Asct2 is localised to the apical side of the renal proximal tubules. Is there any expression of Asct2 in other nephron segments?

To our knowledge, there is no report of ASCT2 expression in other nephron segments. Our immunofluorescent data of the ASCT2 staining in the whole kidney at the low magnification and another region of Figure 3 (below) as well as immunohistochemistry from Human Protein Atlas (update: Jun 9th, 2023) did not show a strong signal of ASCT2 expression in other regions besides the proximal tubules. Thus, we conclude that ASCT2 is mainly expressed in proximal tubules, but not in other nephron regions.

**Author response image 1. sa4fig1:** 

(4) Lines 225-226: Have the authors expressed the candidate genes in HEK293 cells with ASCT2 knockdown?

This experiment was done by expressing the candidate genes in the presence of endogenous ASCT2. We have added the information in lines 225-227 to emphasize this process.

Manuscript lines 225-227

“Based on this finding, we utilized cell growth determination assay as the screening method even in the presence of endogenous ASCT2 expression. HEK293 cells were transfected with human candidate genes without ASCT2 knockdown.”

(5) Lines 254-255: why was D-serine transport enhanced by ASCT2 knockdown in FlpInTRSMCT1 or 2 cells?

We appreciate the reviewer to point out this data. We apologize for causing the confusion in the text. The total amount of D-serine uptake in the cells did not enhance but the net uptake (uptake subtracted from the background) was increased. This enhancement is a result of the lower background by ASCT2 knockdown. We have revised the texts and explained this result in more detail (lines 256-258).

Manuscript lines 256-258

“In the cells with ASCT2 knockdown, the background level was lower, thereby enhancing the D-[3H]serine transport contributed by both SMCT1 and SMCT2 (the net uptake after subtracted with background) (Figure 5C).”

(6) Line 265: The low affinity of SMCT1 for D-serine alone makes it an unlikely transporter for urinary D-serine.

We admitted the reviewer’s concern about the low affinity of SMCT1. However, Km at mM range is widely accepted for several low-affinity amino acid transporters such as proton-coupled amino acid transporter PAT1 (Km = 2 – 5 mM; Miyauchi et al. Biochem J 2010), cationic amino acid transporter CAT2A (Km = 3 – 4 mM; Closs et al. Biochem 1997), and large-neutral amino acid transporter LAT4 (Km = 17 mM; Bodoy et al. J Biol Chem 2005). In the kidneys, many compounds are well-known to be reabsorbed by the low-affinity but high-capacity (high-expression) transporters. Similarly, D-serine was reported to be reabsorbed by the low-affinity transporter (Kragh-Hansen and Sheikh, J Physiol 1984; Shimomura et al. BBA 1988; Silbernagl et al. Am J Physiol Renal Physiol 1999). Moreover, amino acid profile showed urinary D-serine in the range of 100 – 200 µM (Figure supplement 2). This concentration range could drive SMCT1 function (Figure 5). Combined with the high and ubiquitous expression of SMCT1, we propose that SMCT1 is a low-affinity but highcapacity D-serine transporter in the kidneys.

snRNA-seq is a method that can directly compare the expression levels between different genes within the same cells. From Figure supplement 7, expression of SMCT1 is much more abundant than ASCT2. ASCT2 was presented in less than 10% of cell population. It is possible that 90% of the cells that do not express ASCT2 use SMCT1 to reabsorb Dserine.

We have revised the Discussion regarding this comment (lines 386-404).

Manuscript lines 386-404

“Kinetics analysis of D-serine transport revealed the high affinity by ASCT2 (Km 167 µM) (Foster et al., 2016) and low affinity by SMCT1 (Km 3.39 mM; Figure 5E). In addition to transport affinity, the expression levels and co-localization of multiple transporters within the same cells are critical for elucidating the physiological roles of transporters or transport systems (Sakaguchi et al., 2024). In our proteome data, the chromatogram intensities of Smct1 (2.9 x 109 AU) and Smct2 (1.6 x 108 AU) were significantly higher than that of Asct2 (1.5 x 107 AU) in the control mice (Table 1: abundance in Sham). While direct intensity comparisons between different proteins in mass spectrometry analyses are not precise, they can provide a general indication of relative protein amounts. This finding aligns with the snRNA-seq data, where Asct2 expression was found to be minimal, present in less than 10% of cell populations under both control and IRI conditions, suggesting that many cells do not express Asct2. Conversely, Smct1 and Smct2 show high and ubiquitous expression in control conditions, but their levels are markedly reduced in IRI conditions (Figure supplement 7). Our ex vivo assays demonstrate that both ASCT2 and SMCTs mediate D-serine transport (Figure 7B). Consequently, Asct2 may contribute to D-serine reabsorption due to its high affinity, whereas Smcts, owing to their abundance, particularly in cells lacking Asct2, likely play a significant role in D-serine reabsorption. Moreover, factors such as transport turnover rate (Kcat) and the presence of local canonical substrates are also vital in defining the overall contribution of Dserine transport systems.”

(7) Line 316: The authors state that there is a high tubular D-serine reabsorption in IRI and in line 424 there is an inactivation of DAAO during the pathology. This suggests that there is a reabsorption of D-serine mediated by a transport system in the basolateral membrane domain of proximal tubular cells. Do the authors have any information about this transporter?

We agree with the reviewer that transporters at the basolateral membrane are important to complete the D-serine reabsorption in the kidney, and have included this issue in the original manuscript. We stated that transport systems at the basolateral side are necessary to be analyzed in order to complete the picture of D-serine transport systems in the kidney (lines 481-483 of the revised manuscript). However, we did not have any strong candidates for basolateral D-serine transport systems. Because we analyzed the proteome of BBMV, which concentrates on the apical membrane proteins, the analysis did not detect several transporters at the basolateral side.

(8) In lines 462-463, the authors state: "It is suggested that PAT1 is less active at the apical membrane where the luminal pH is neutral". However, the pH of urine in the proximal tubules is normally acidic due to the high activity of NH3. I suggest rewording this sentence.

Thank you for your comment. Proximal tubule (PT) is the first and the main region to maintain acid-base homeostasis in the kidney. In PT cells, NH3 secretes H+ to titrate luminal HCO3- and creates CO2, which is absorbed into PT cells and produces "new intracellular HCO3-", which is subsequently reabsorbed into the blood. Although ion fluxes in PT is to maintain the pH homeostasis, the pH regulation in both luminal and intracellular PT cells is highly dynamic. We totally agree with the reviewer and to follow that, we have revised the text by emphasizing the pH around PT segments, rather than the final urine pH, and leaving the discussion open for the possibility of PAT1 function in PT of normal kidneys (lines 474481).

Manuscript lines 474-481

“PAT1, a low-affinity proton-coupled amino acid transporter (Km in mM range), has been found at both sub-apical membranes of the S1 segment and inside of the epithelia (The Human Protein Atlas: https://www.proteinatlas.org; updated on Dec 7th, 2022) (Sagné et al., 2001; Vanslambrouck et al., 2010). PAT1 exhibits optimum function at pH 5 - 6 but very low activity at pH 7 (Miyauchi et al., 2005; Bröer, 2008b). Future research is required to address the significance of PAT1 on D-serine transport in the proximal tubule segments where pH regulation is known to be highly dynamic (Boron, 2006; Nakanishi et al., 2012; Bouchard and Mehta, 2022; Imenez Silva and Mohebbi, 2022).”

**Reviewer #3 (Recommendations For The Authors):**
The authors proposed that the increased expression of ASCT2, even together with the decreased expression of SMCT1/2, causes the increased renal reabsorption of D-serine that occurs in IRI. In the discussion, the main argument to sustain this hypothesis is the higher apparent affinity for D-serine of ASCT2 (<200 uM Km) versus SMCT1 (3.4 mM Km). In the Discussion section (page 18- 1st complete paragraph), the authors indicate that the Mass Spec intensities of SMCT1 and 2 are two and one order of magnitude higher respectively than that of ASCT2. This suggests that SMCT1 is clearly more expressed than ASCT2 in control conditions. IRI increments ASCT2 protein expression in brush-border membrane vesicle from kidney 1.6 folds and decreases that of SMCT1 0.6 folds. How this fold changes, even taking into account the lower Km of ASCT2 versus SMCT1 would explain the dramatic changes in the D-/L-serine ratios in plasma and urine in IRI? The authors might discuss whether other transport characteristics, even unknown (e.g., a higher turnover rate of ASCT2 vs SMCT1), would also contribute to the higher D-serine reabsorption in IRI.SMCT1 shows some enantiomer selectivity for D- vs L-serine. At 50 uM concentration the transport is almost double for D. vs L-serine, but is ASCT2 stereoselective between the two enantiomers of serine? Some of the authors of this manuscript showed in a previous paper that the basolateral transporter Asc1 also participates in the accumulation of D-serine in serum caused by renal tubular damage. (Serum D-serine accumulation after proximal renal tubular damage involves neutral amino acid transporter Asc-1. Suzuki M et al. Sci Rep. 2019 Nov 13;9(1):16705 (PMID: 31723194)). Asc1 shows no stereoselectivity between L- and D-serine. Can the authors discuss possible mechanisms resulting in increased renal reabsorption of Dserine than L-serine in IRI with the participation of transporters with modest stereoselectivity for D- vs L-serine?

We appreciate the reviewer’s comments on the degree of protein alteration in proteomics, the functional contributions of ASCT2 and SMCTs, and the alteration of D/L ratios. We have included the possibilities of the technical concerns and the discussion on the roles of ASCT2 and SMCTs as follows.

Regarding the expression levels, proteomics and snRNA-seq showed the same tendency that ASCT2 increase and SMCTs decrease in IRI conditions. However, the degrees of alterations are more contrast in snRNA-seq. This may be due to the difference in quantification methods and probably points out the underestimated quantification of membrane transport proteins in label-free proteomics. The accuracy of protein quantifications in the label-free proteomics are often impacted by the presence of post-translational modifications and multiple trans-membrane domains like in the case of the membrane transport proteins (Schey KL et al. Biochemistry 2015, doi: 10.1021/bi301604j). Alternative methods of quantitative proteomics may be added in the future for a more thorough comparison. We have added this issue in lines 351-356 of the revised version.

Manuscript lines 351-356

“When evaluating the extent of gene/protein alterations between the control and IRI conditions, we observed that the gene alterations of both Asct2 and Smcts, as revealed by snRNA-sequencing, are more pronounced than the protein alteration ratios obtained from proteomics. This discrepancy may stem from difficulty in the quantification method, especially for membrane transport proteins in label-free quantitative proteomics.”

For the functional contributions of ASCT2 and SMCTs in the kidney, we admitted the reviewer’s concern about the low affinity of SMCT1. Following the reviewer’s comment, we have included other factors besides transport affinities, e.g. expression levels and turnover rates of the transporters. From the results of both proteomics and snRNA-seq, ASCT2 expression is significantly lower than SMCTs in the normal conditions. snRNA-seq showed that ASCT2 was presented in less than 10% of the cell population (Figure supplement 7). We propose that most of the cells that do not express ASCT2 may use SMCT1 to reabsorb D-serine. This topic was included in the revised manuscript lines 386-404.

Manuscript lines 386-404

“Kinetics analysis of D-serine transport revealed the high affinity by ASCT2 (Km 167 µM) (Foster et al., 2016) and low affinity by SMCT1 (Km 3.39 mM; Figure 5E). In addition to transport affinity, the expression levels and co-localization of multiple transporters within the same cells are critical for elucidating the physiological roles of transporters or transport systems (Sakaguchi et al., 2024). In our proteome data, the chromatogram intensities of Smct1 (2.9 x 109 AU) and Smct2 (1.6 x 108 AU) were significantly higher than that of Asct2 (1.5 x 107 AU) in the control mice (Table 1: abundance in Sham). While direct intensity comparisons between different proteins in mass spectrometry analyses are not precise, they can provide a general indication of relative protein amounts. This finding aligns with the snRNA-seq data, where Asct2 expression was found to be minimal, present in less than 10% of cell populations under both control and IRI conditions, suggesting that many cells do not express Asct2. Conversely, Smct1 and Smct2 show high and ubiquitous expression in control conditions, but their levels are markedly reduced in IRI conditions (Figure supplement 7). Our ex vivo assays demonstrate that both ASCT2 and SMCTs mediate D-serine transport (Figure 7B). Consequently, Asct2 may contribute to D-serine reabsorption due to its high affinity, whereas Smcts, owing to their abundance, particularly in cells lacking Asct2, likely play a significant role in D-serine reabsorption. Moreover, factors such as transport turnover rate (Kcat) and the presence of local canonical substrates are also vital in defining the overall contribution of D-serine transport systems.”

As for the dramatic alterations of D/L-serine ratios juxtaposed with minimal changes in ASCT2 and SMCTs expression level, we cautiously refrain from drawing a definitive conclusion regarding the entire mechanism. This caution is grounded in the scientific understanding of a comprehensive elucidation of both L-serine transport systems and D-serine transport systems at both apical and basolateral membranes. Nevertheless, we would like to suggest a mechanism at the apical membrane based on the current knowledge.

For D-serine transport systems, we found ASCT2 and SMCTs contributions in this study. Meanwhile, L-serine was previously reported to be mediated mainly by the neutral amino acid transporters B0AT3 (in particular B0AT3; Bröer et al. Physiol Rev 2008; Singer et al. J Biol Chem 2009). Hence, the mechanism behind the alterations of D/L-serine ratios should include B0AT3 functions as well. In IRI conditions, B0AT3 decreased 1.8 fold. We suggest that high ratios of D-/L-serine in IRI conditions are a combined outcome of (1) increase of D-serine reabsorption by ASCT2 enhancement and SMCTs reduction, and (2) decrease of L-serine reabsorption by B0AT3. We have included this suggestion in the Discussion lines 438-451.

Manuscript lines 438-451

“The enantiomeric profiles of serine revealed distinct plasma D/L-serine ratios, with low ratios in the normal control but elevated ratios in IRI, despite the weak stereoselectivity of ASCT2 (Figure 1B). This observation suggested the differential renal handling of D-serine compared to L-serine. While we identified SMCTs as a Dserine transport system, it has been reported that L-serine reabsorption is mediated by B0AT3 (Singer et al., 2009). We propose that the alterations in plasma and urinary D/Lserine ratios are the combined outcomes of: (1) transport systems for L-serine, and (2) transport systems for D-serine. In normal kidneys, the low plasma D/L-serine ratios could result from the efficient reabsorption of L-serine by B0AT3, coupled with the DAAO activity that degrades intracellular D-serine reabsorbed by SMCTs. In IRI conditions, our enantiomeric amino acid profiling revealed low plasma L-serine and high urinary L-serine (Figure supplements 1B, 2B). Additionally, the proteomics analysis indicated a reduction in B0AT3 levels (4h IRI/sham = 0.56 fold; 8h IRI/sham = 0.65 fold; Table S1). These observations suggest that the low L-serine reabsorption in IRI is a result of B0AT3 reduction.”

In the case of Asc-1, it was reported to be a D-serine transporter in the brain (Rosenberg et al. J Neurosci 2013). Suzuki et al. 2019 showed the increase of Asc-1 in cisplatin-induced tubular injury. Notably, the mRNA of Asc-1 is predominantly found in Henle’s loop, distal tubules, and collecting ducts but not in proximal tubules, and its protein expression level is dramatically low in the kidney (Human Protein Atlas: update on Jun 19, 2023). Furthermore, in this study, Asc-1 expression was not detected in the brush border membrane proteome. Consequently, we have decided not to include Asc-1 in the Discussion of this study, which primarily focuses on the proximal tubules.